# FEASIBLE POLICY OPTIMIZATION FOR SAFE REINFORCEMENT LEARNING

## ABSTRACT

Policy gradient methods serve as a cornerstone of reinforcement learning (RL), yet their extension to safe RL, where policies must strictly satisfy safety constraints, remains challenging. While existing methods enforce constraints in every policy update, we demonstrate that this is unnecessarily conservative. Instead, each update only needs to progressively expand the feasible region while improving the value function. Our proposed algorithm, namely feasible policy optimization (FPO), simultaneously achieves both objectives by solving a region-wise policy optimization problem. Specifically, FPO maximizes the value function inside the feasible region and minimizes the feasibility function outside it. We prove that these two sub-problems share a common optimal solution, which is obtained based on a tight bound we derive on the constraint decay function. Extensive experiments on the Safety-Gymnasium benchmark show that FPO achieves excellent constraint satisfaction while maintaining competitive task performance, striking a favorable balance between safety and return compared to state-of-the-art safe RL algorithms.

## 1 INTRODUCTION

Reinforcement learning (RL) has demonstrated remarkable success in domains ranging from board games (Schrittwieser et al., 2020) and racing simulations (Wurman et al., 2022) to recent breakthroughs in large language models (Guo et al., 2025). Despite these successes, a fundamental challenge persists: current methods primarily excel in simulated environments where unsafe behaviors carry no real cost, while in safety-critical applications, policy failures could lead to severe consequences. Addresses this challenge requires considering a constrained optimal control problem, where policies must strictly satisfy safety constraints at all times, also known as state-wise constraints (Zhao et al., 2023b), while maximizing expected returns (Yang et al., 2024).

Policy gradient (PG) is a foundational method in RL (Li, 2023), which formulates RL as an optimization problem and applies gradient-based methods to solve it. This framework has given rise to powerful modern deep RL algorithms such as proximal policy gradient (PPO) (Schulman et al., 2017) and group relative policy optimization (GRPO) (Shao et al., 2024). However, a critical limitation of standard PG methods is that they are not directly applicable to safe RL because of their unconstrained problem formulation. Despite many well-established constrained optimization techniques (Boyd & Vandenberghe, 2004), integrating them with PG while maintaining high training efficiency remains an open challenge.

Existing safe RL methods fall into two categories. A prominent class is called iterative unconstrained RL, which reformulates safe RL as a sequence of unconstrained optimization problems, typically via the method of Lagrange multipliers, and solves them using standard RL algorithms (Paternain et al., 2019). While theoretically sound, these methods suffer from slow convergence and training instability. Slow convergence arises from the need to solve an RL problem in each iteration, resulting in convergence rates approximately an order of magnitude slower than standard RL algorithms. Training instability stems from the characteristic of the Lagrange multiplier, manifesting as persistent oscillations in return and constraint violation throughout training (Stooke et al., 2020).

Another class of methods, called constrained policy optimization, aligns more closely with PG, or more generally, policy optimization, which employs more advanced optimization techniques than

pure gradient ascent. These methods impose the safety constraint on the sub-problem in each iteration, requiring every intermediate policy to be strictly safe. While more efficient than unconstrained iterative RL, these methods suffer from the infeasibility issue: they often fail to find a constraint-satisfying solution to the sub-problems, especially during early training stages. This is because the constraint is too stringent for policies that have not been sufficiently trained after random initialization. In such cases, these methods must resort to pure constraint minimization without reward optimization (Achiam et al., 2017), resulting in overly conservative updates and inefficient training.

In this paper, we challenge the conventional practice of enforcing the original constraint in every iteration of policy optimization. Instead, we demonstrate that each iteration only needs to progressively expand the feasible region while improving the value function. This insight is theoretically grounded in feasible policy iteration (FPI) (Yang et al., 2023c), which proves that such updates guarantee convergence to the maximum feasible region and the optimal value function. Our approach replaces the stringent constraint that every policy must be strictly safe with a milder one: each policy only needs to be safer than the previous one in the sense that its feasible region is expanded. Building on this foundation, we propose feasible policy optimization (FPO), which maximizes the value function inside the feasible region and minimizes the feasibility function outside it. We prove that these two objectives, originally expressed by two separate optimization problems, can be simultaneously achieved with a shared optimal solution. We further derive a tight bound on the constraint decay function (CDF), enabling more accurate feasible region estimation compared to the conventional cost value function (CVF). Extensive evaluation on the Safety-Gymnasium benchmark demonstrates FPO's excellent balance between safety and return.

## 2 RELATED WORK

**Iterative unconstrained RL**   Most iterative unconstrained RL methods use the method of Lagrange multipliers and solve the dual problem using dual ascent, where the minimization step solves an unconstrained RL problem (Paternain et al., 2019). For example, Chow et al. (2018) constrain the conditional value-at-risk of the CVF in a constrained Markov decision process (Altman, 2021), forming a probabilistic constraint. Tessler et al. (2018) incorporate the cost signal into the reward function, treating the integrated discounted sum as a new value function. The Lagrange multiplier framework is also adaptable to other kinds of feasibility functions, including Hamilton-Jacobi reachability (Yu et al., 2022; 2023), control barrier function (Yang et al., 2023a;b), and safety index (Ma et al., 2022). As a special case, when the multiplier is fixed as a constant, the algorithm reduces to a penalty function method (Thomas et al., 2021).

**Constrained policy optimization**   The most representative example of this class is the constrained policy optimization (CPO) algorithm (Achiam et al., 2017), which builds on the trust region policy optimization (TRPO) (Schulman et al., 2015) and further adds a linearized safety constraint. To avoid the computationally expensive line search in CPO, Yang et al. (2020) propose to first perform a reward improvement update and then project the policy back onto the constrained set. Zhang et al. (2020) propose to first solve for the optimal policy in a non-parameterized policy space and then project it back into the parametric space. Following the projection method, Yang et al. (2022) propose generalized advantage estimation (GAE) for the surrogate function to further improve performance. Inspired by techniques from constrained optimization, the interior-point method (Liu et al., 2020) and the augmented Lagrange method (Dai et al., 2023) are also explored to solve the policy optimization problem in each iteration. For finite-horizon problems, Zhao et al. (2023a) and Zhao et al. (2024) convert state-wise constraints to cumulative constraints through cost reconstruction and bound the worst-case violation.

## 3 PRELIMINARIES

### 3.1 PROBLEM STATEMENT

Safe RL addresses control problems in which an agent aims to maximize long-term rewards while strictly adhering to safety constraints at every step. We consider a Markov decision process (MDP) $(\mathcal{X}, \mathcal{U}, d_{\text{init}}, P, r, \gamma)$, where $\mathcal{X} \subseteq \mathbb{R}^n$ is the state space, $\mathcal{U} \subseteq \mathbb{R}^m$ is the action space, $d_{\text{init}} \in \Delta\mathcal{X}$ is the initial state distribution, $P : \mathcal{X} \times \mathcal{U} \to \Delta\mathcal{X}$ is the transition probability, $r : \mathcal{X} \times \mathcal{U} \to \mathbb{R}$ is the reward

function, and $0 < \gamma < 1$ is the discount factor. We consider a stochastic policy $\pi : \mathcal{X} \to \Delta\mathcal{U}$, whose value function is defined as:

$$V^\pi(x) = \mathbb{E}_{x_{t+1} \sim P(\cdot|x_t, u_t), u_t \sim \pi(\cdot|x_t)} \left[ \sum_{t=0}^{\infty} \gamma^t r(x_t, u_t) \Big| x_0 = x \right]. \tag{1}$$

Safety is specified through a state constraint expressed as an inequality $h(x) < 0$, where $h : \mathcal{X} \to \mathbb{R}$ is the constraint function. We aim to find a policy that maximizes the expected value function while satisfying the state constraint at every step over an infinite horizon:

$$\max_\pi \mathbb{E}_{x \sim d_{\text{init}}}[V^\pi(x)] \tag{2}$$
$$\text{s.t. } h(x_t) \leq 0, \forall t \geq 0, x_0 \in \mathcal{X}_{\text{init}},$$

where $\mathcal{X}_{\text{init}} = \{x \in \mathcal{X} | d_{\text{init}}(x) > 0\}$ is the support of the initial state distribution.

## 3.2 FEASIBLE REGION AND FEASIBILITY FUNCTION

The constrained optimal control problem (2) is intractable because it has infinitely many constraints. A common solution is to aggregate these constraints into a single one through a feasibility function. To formally describe the concept of feasibility, we first define the reachable set.

**Definition 1** (Reachable set). *The reachable set of a policy $\pi$ from a state $x \in \mathcal{X}$, denoted $\mathcal{R}^\pi(x)$, is the set of states that can be reached with non-zero probability under $\pi$ in finite time:*

$$\mathcal{R}^\pi(x) = \{x' \in \mathcal{X} | \exists t \geq 0, \text{ s.t. } P(x_t = x'|x, \pi) > 0\}, \tag{3}$$

*where $P(x_t = x'|x, \pi)$ is the probability of reaching $x'$ at time $t$ starting from $x$ and following $\pi$.*

We call a state feasible under a policy if all its future states satisfy the safety constraint, and the set of all feasible states under a policy is the feasible region of the policy.

**Definition 2** (Feasible region). *The feasible region of a policy $\pi$, denoted $\mathrm{X}^\pi$, is the set of states from which every reachable set under $\pi$ satisfies the safety constraint:*

$$\mathrm{X}^\pi = \{x \in \mathcal{X} | \forall x' \in \mathcal{R}^\pi(x), h(x') \leq 0\}. \tag{4}$$

The feasible region enables us to describe the long-term safety requirement compactly: the feasible region must include all possible initial states. This requirement can be expressed as a single constraint by the feasibility function.

**Definition 3** (Feasibility function). *Function $F^\pi : \mathcal{X} \to \mathbb{R}$ is a feasibility function of $\pi$ if and only if its zero-sublevel set equals the feasible region of $\pi$, i.e., $\{x \in \mathcal{X} | F^\pi(x) \leq 0\} = \mathrm{X}^\pi$.*

An example of a feasibility function is the CDF (Yang et al., 2023b).

**Definition 4** (Constraint decay function). *The CDF of a policy $\pi$ is defined as*

$$F^\pi(x) = \mathbb{E}_{\tau \sim \pi} \left[ \gamma^{N(\tau)} \Big| x_0 = x \right], \tag{5}$$

*where $\gamma \in (0, 1)$ is the discount factor, $\tau = \{x_0, u_0, x_1, u_1, \dots\}$ is a trajectory sampled by $\pi$, and $N(\tau) \in \mathbb{N}$ is the time step of the first constraint violation in $\tau$.*

The CDF is non-negative by definition, and thus its zero-sublevel set equals its zero-level set. Without loss of generality, we only consider non-negative feasibility functions in this paper. For feasibility functions with negative values, we can take their non-negative parts $F_+^\pi = \max\{F^\pi, 0\}$ without changing the feasible region. With a feasibility function, we can aggregate the infinitely many constraints in Problem (2) into a single one, obtaining the following problem:

$$\max_\pi \mathbb{E}_{x \sim d_{\text{init}}}[V^\pi(x)] \quad \text{s.t. } \mathbb{E}_{x \sim d_{\text{init}}}[F^\pi(x)] \leq 0. \tag{6}$$

## 4 METHODS

Existing constrained policy optimization methods typically require that every intermediate policy satisfies the constraint in Problem (6). Instead, our algorithm only requires each policy to have a larger feasible region than the previous policy, which can be achieved through a region-wise policy optimization scheme.

### 4.1 REGION-WISE POLICY OPTIMIZATION

We propose to solve two optimization problems in each iteration. Let $\pi_k$ denote the policy from the previous iteration. The first problem is to maximize the value function inside the feasible region under the constraint that the new feasible region is not smaller:

$$\max_{\pi} \mathbb{E}_{x \sim d_{\text{init}}}[\mathbb{I}[F^{\pi_k}(x) \leq 0]V^{\pi}(x)]$$
$$\text{s.t. } \mathbb{E}_{x \sim d_{\text{init}}}[\mathbb{I}[F^{\pi_k}(x) \leq 0]F^{\pi}(x)] \leq 0. \tag{7}$$

The second problem is to minimize the feasibility function outside the feasible region under the same constraint:

$$\min_{\pi} \mathbb{E}_{x \sim d_{\text{init}}}[\mathbb{I}[F^{\pi_k}(x) > 0]F^{\pi}(x)]$$
$$\text{s.t. } \mathbb{E}_{x \sim d_{\text{init}}}[\mathbb{I}[F^{\pi_k}(x) \leq 0]F^{\pi}(x)] \leq 0. \tag{8}$$

The next policy $\pi_{k+1}$ is obtained by solving Problem (7) and (8), which, we will prove, have a shared optimal solution. The theoretical basis of this policy update rule is provided by FPI (Yang et al., 2023c), which proves that in finite state and action spaces, this update rule produces monotonically improved value functions and feasible regions, with guaranteed convergence to the optimal solution to the original safe RL problem (6). We generalize the update rule of FPI to infinite spaces by replacing the state-wise optimization with expectation optimization.

**Theorem 1.** *There exists a policy $\pi_{k+1}$ that is the optimal solution to both Problem (7) and (8).*

*Proof Sketch.* Let $\pi_{\text{in}}$ and $\pi_{\text{out}}$ denote the optimal solutions to Problem (7) and (8), respectively. We construct the following policy:

$$\pi_{k+1}(\cdot|x) = \begin{cases} \pi_{\text{in}}(\cdot|x), & x \in \mathcal{R}^{\pi_{\text{in}}}(\mathcal{X}_{\text{init}} \cap X^{\pi_k}), \\ \pi_{\text{out}}(\cdot|x), & \text{otherwise}, \end{cases} \tag{9}$$

where $\mathcal{R}^{\pi}(X) = \bigcup_{x \in X} \mathcal{R}^{\pi}(x)$ denotes the reachable set of $\pi$ from a set of states $X \subseteq \mathcal{X}$. We prove that $\pi_{k+1}$ is the optimal solution to both problems. The key is to observe that $\mathcal{R}^{\pi_{\text{in}}}(\mathcal{X}_{\text{init}} \cap X^{\pi_k})$ is forward invariant under $\pi_{k+1}$. See Appendix A.1 for the complete proof. $\square$

Note that Equation (9) only provides one valid choice of $\pi_{k+1}$. There may exist other valid policies, such as remaining with $\pi_k$ in the overlapping part of the reachable sets. Theorem (1) allows us to merge Problem (7) and (8) into a single problem as follows:

$$\max_{\pi} \mathbb{E}_{x \sim d_{\text{init}}}[\mathbb{I}[F^{\pi_k}(x) \leq 0]V^{\pi}(x) - \mathbb{I}[F^{\pi_k}(x) > 0]F^{\pi}(x)]$$
$$\text{s.t. } \mathbb{E}_{x \sim d_{\text{init}}}[\mathbb{I}[F^{\pi_k}(x) \leq 0]F^{\pi}(x)] \leq 0. \tag{10}$$

**Corollary 1.** *The optimal solution to Problem (10) is also the optimal solution to both Problem (7) and (8).*

This is because the objective function of Problem (10) is the sum of the objective functions of Problem (7) and (8), and they share the same constraint. Thus, $\pi_{k+1}$ defined in (9) is the optimal solution to all three problems.

### 4.2 FEASIBILITY FUNCTION BOUNDS

A difficulty of solving Problem (10) is that the value function and feasibility function of the new policy $\pi$ cannot be directly approximated with samples collected by the old policy $\pi_k$. To solve this problem, we replace the two functions with their lower and upper bounds, which can be approximated by samples from the old policy. Achiam et al. (2017) derive the bounds for functions in the form of discounted summation, which is applicable to the value function. In this section, we move a step further and derive the bounds for CDF.

We begin with a decomposition of state distribution. Given an initial state $x \in \mathcal{X}$, the discounted future state distribution under policy $\pi$ is $d^{\pi}(x'|x) = (1 - \gamma)\sum_{t=0}^{\infty} \gamma^t P(x_t = x'|x, \pi)$. By law of total probability, we decompose each term in the summation based on whether the constraint has been violated up to that step:

$$P(x_t = x'|x, \pi) = P(x_t = x', \max_{s < t} c_s = 0|x, \pi) + P(x_t = x', \max_{s < t} c_s = 1|x, \pi),$$

where $c_s = \mathbb{I}[h(x_s) > 0]$ is the indicator function for constraint violation. Then, the future state distribution can be decomposed as $d^\pi(x'|x) = d_0^\pi(x'|x) + d_+^\pi(x'|x)$, where

$$d_0^\pi(x'|x) = (1 - \gamma) \sum_{t=0}^\infty \gamma^t P(x_t = x', \max_{s<t} c_s = 0 | x, \pi),$$

$$d_+^\pi(x'|x) = (1 - \gamma) \sum_{t=0}^\infty \gamma^t P(x_t = x', \max_{s<t} c_s = 1 | x, \pi).$$

We call $d_0^\pi$ the prefix state distribution. This decomposition is critical in deriving the bounds for CDF. As we will show later, the bounds for CDF only depend on the prefix state distribution because states beyond the first violation are irrelevant to the CDF. In the following analysis, we slightly abuse notation by writing $\mathbb{E}_{x \sim d_0^\pi}[f(x)]$ to represent $\int_{\mathcal{X}} f(x) d_0^\pi(x) \mathrm{d}x$ even when $\int_{\mathcal{X}} d_0^\pi(x) \mathrm{d}x < 1$.

**Theorem 2.** *For any policies $\tilde{\pi}$ and $\pi$, and any state $x \in \mathcal{X}$, define*

$$A_F^\pi(x, u) = \mathbb{E}_{x' \sim P(\cdot|x,u)}[c(x) + (1 - c(x))\gamma F^\pi(x') - F^\pi(x)],$$

*and $L_{\tilde{\pi}}^\pi(x) = \mathbb{E}_{x' \sim d_0^\pi(\cdot|x), u' \sim \tilde{\pi}(\cdot|x')}[A_F^\pi(x', u')]$, $\epsilon_F^{\tilde{\pi}} = \max_x |\mathbb{E}_{u \sim \tilde{\pi}(\cdot|x)}[A_F^\pi(x, u)]|$. Then,*

$$F^{\tilde{\pi}}(x) - F^\pi(x) \geq \frac{L_{\tilde{\pi}}^\pi(x)}{1 - \gamma} - \frac{2\gamma \epsilon_F^{\tilde{\pi}}}{(1 - \gamma)^2} \mathbb{E}_{x' \sim d_0^\pi(\cdot|x)}[D_{TV}(\tilde{\pi}\|\pi)[x']],$$

$$F^{\tilde{\pi}}(x) - F^\pi(x) \leq \frac{L_{\tilde{\pi}}^\pi(x)}{1 - \gamma} + \frac{2\gamma \epsilon_F^{\tilde{\pi}}}{(1 - \gamma)^2} \mathbb{E}_{x' \sim d_0^\pi(\cdot|x)}[D_{TV}(\tilde{\pi}\|\pi)[x']],$$

*where $D_{TV}(\tilde{\pi}\|\pi)[x'] = (1/2) \sum_u |\tilde{\pi}(u|x') - \pi(u|x')|$ is the total variational divergence between action distributions at $x'$. Furthermore, the bounds are tight (when $\tilde{\pi} = \pi$, the LHS and RHS are identically zero).*

*Proof Sketch.* We construct an auxiliary MDP $\tilde{M}$, which is identical to $M$ except for its transition probability. In $\tilde{M}$, once the constraint is violated, the state is fixed at the one that violates the constraint for all future steps. We prove that the CDF and prefix state distribution are identical in $M$ and $\tilde{M}$, and the result to prove holds in $\tilde{M}$. Therefore, the result also holds in $M$. See Appendix A.2 for the complete proof. $\qquad\square$

One may ask why not use the CVF as the feasibility function, which is a discounted summation so that the bound from CPO would still apply. The reason is that CDF yields more accurate estimates than CVF in practice. In safe RL, feasibility functions are typically estimated using bootstrapping methods like TD($\lambda$), which suffer from approximation bias of the feasibility function itself. While this bias affects both CDF and CVF, CVF suffers more severely because it is unbound and requires infinite-horizon trajectories. In contrast, CDF is bounded within $[0, 1]$, allowing the bootstrapping target to be explicitly clipped, and its shorter trajectories (truncated at first violation) also decrease variance.

## 4.3 FEASIBLE POLICY OPTIMIZATION

With the CDF bounds, we are ready to solve Problem (10). Substituting the upper bound of CDF from Theorem 2 and the lower bound of value function from Corollary 1 in the CPO paper (Achiam et al., 2017), and following the practice of trust region methods, we obtain the following problem:

$$\max_\pi \mathbb{E}_{x \sim d^{\pi_k}, u \sim \pi} \left[\mathbb{I}[F^{\pi_k}(x) \leq 0] A^{\pi_k}(x, u)\right] - \mathbb{E}_{x \sim d_0^{\pi_k}, u \sim \pi} \left[\mathbb{I}[F^{\pi_k}(x) > 0] A_F^{\pi_k}(x, u)\right]$$

$$\text{s.t. } \mathbb{E}_{x \sim d^{\pi_k}, u \sim \pi} \left[\mathbb{I}[F^{\pi_k}(x) \leq 0](F^{\pi_k}(x) + A_F^{\pi_k}(x, u)/(1 - \gamma))_+\right] \leq 0 \tag{11}$$

$$\mathbb{E}_{x \sim d^{\pi_k}} \left[\mathbb{I}[F^{\pi_k}(x) \leq 0] D_{KL}(\pi\|\pi_k)[x]\right] \leq \delta/2$$

$$\mathbb{E}_{x \sim d^{\pi_k}} \left[\mathbb{I}[F^{\pi_k}(x) > 0] D_{KL}(\pi\|\pi_k)[x]\right] \leq \delta/2.$$

Here, $A^{\pi_k}(x, u) = Q^{\pi_k}(x, u) - V^{\pi_k}(x)$ is the standard advantage function in RL. In the above constraints, we replace the prefix state distribution $d_0^\pi$ with the whole state distribution $d^\pi$. This replacement is valid because $d^\pi \geq d_0^\pi$ for all states. Our algorithm, called feasible policy optimization (FPO), iteratively solves Problem (11) to update the policy. This update rule provides the following guarantees on the safety and performance of the new policy.

**Corollary 2.** *The optimal solution to Problem (11), denoted $\pi_{k+1}$, satisfies the following two properties:*

*1. Feasibility enhancement:*

$$\mathbb{E}_{x \sim d_{init}} \left[ \mathbb{I}[F^{\pi_k}(x) \leq 0] F^{\pi_{k+1}}(x) \right] \leq \frac{\sqrt{\delta} \gamma \epsilon_F^{\pi_{k+1}}}{(1-\gamma)^2}, \tag{12a}$$

$$\mathbb{E}_{x \sim d_{init}} \left[ \mathbb{I}[F^{\pi_k}(x) > 0] \left( F^{\pi_{k+1}}(x) - F^{\pi_k}(x) \right) \right] \leq \frac{\sqrt{\delta} \gamma \epsilon_F^{\pi_{k+1}}}{(1-\gamma)^2}, \tag{12b}$$

*where $\epsilon_F^{\pi_{k+1}} = \max_x |\mathbb{E}_{u \sim \pi_{k+1}(\cdot|x)}[A_F^\pi(x, u)]|$.*

*2. Value improvement:*

$$\mathbb{E}_{x \sim d_{init}} \left[ \mathbb{I}[F^{\pi_k}(x) \leq 0] \left( V^{\pi_{k+1}}(x) - V^{\pi_k}(x) \right) \right] \geq -\frac{\sqrt{\delta} \gamma \epsilon^{\pi_{k+1}}}{(1-\gamma)^2}, \tag{13}$$

*where $\epsilon^{\pi_{k+1}} = \max_x |\mathbb{E}_{u \sim \pi_{k+1}(\cdot|x)}[A^\pi(x, u)]|$.*

*Proof Sketch.* Split (11) into two problems similar to Section 4.1 and prove that they share the same optimal solution. The rest follows by Theorem 2 and Corollary 1 in the CPO paper (Achiam et al., 2017). See Appendix A.3 for the complete proof. □

This corollary tells us that the safety and performance degradation of the new policy is controlled. Specifically, its feasibility function will not exceed zero too much inside the feasible region or increase too much outside the feasible region, and its value function will not decrease too much inside the feasible region. As the step size $\delta$ decreases, the policy sequence obtained by FPO approaches a monotonically improving sequence in both safety and performance.

### 4.4 PRACTICAL IMPLEMENTATION

We adopt the method from PPO to solve Problem (11), which applies a first-order method with the KL divergence constraints replaced by a clipped importance sampling (IS) ratio. FPO learns a feasibility network $F_\phi$, a value network $V_\omega$, and a policy network $\pi_\theta$, where $\phi$, $\omega$, and $\theta$ denote their parameters. We additionally introduce a hyperparameter $\epsilon > 0$ and approximate feasibility by $F_\phi(x) \leq \epsilon$. This is because, in practice, approximation error causes the CDF to be positive almost everywhere since its learning target is non-negative. This approximation is valid under the assumption that the step to violation is uniformly bounded (Thomas et al., 2021). In our experiments, we find that a fixed value of $\epsilon = 0.1$ works well for all environments.

We deal with the constraint inside the feasible region by penalizing the advantage function. Specifically, we take a weighted sum of the reward advantage and feasibility advantage:

$$\bar{A}(x, u) = \mathbb{I}[F_\phi(x) \leq \epsilon](\alpha(x) A(x, u) + (1 - \alpha(x)) A_F(x, u)) + \mathbb{I}[F_\phi(x) > \epsilon] A_F(x, u),$$

where the weight $\alpha(x) = (1 - F_\phi(x)/\epsilon)^\beta$, and the exponent $\beta > 0$ is updated by

$$\beta \leftarrow \beta + \eta \mathbb{E}_{x \sim d^{\pi_{\theta_k}}, u \sim \pi_\theta} \left[ \mathbb{I}[F_\phi(x) \leq \epsilon](F_\phi(x) + A_F(x, u)/(1-\gamma) - \epsilon)_+ \right], \tag{14}$$

where $\eta$ is the learning rate. The reason for designing the weight in this way is that states with CDF values close to $\epsilon$ are more likely to become infeasible after an update step. Thus, we need to put more weight on the feasibility advantage of these states to prevent them from becoming infeasible. To compute the feasibility advantage, we extend the GAE of the value function to the CDF:

$$A_F(x, u) = \sum_{t=0}^{\infty} (\lambda \gamma)^t \prod_{s=0}^{t-1} (1 - c_s) \left( c_t + (1 - c_t) \gamma F_\phi(x_{t+1}) - F_\phi(x_t) \right). \tag{15}$$

See Appendix B.1 for the detailed derivation.

The loss function for the feasibility network is

$$L_F(\phi) = \mathbb{E} \left[ \left( F_\phi(x) - (F_{\phi_k}(x) + A_F(x, u)) \right)^2 \right]. \tag{16}$$

The loss function for the value network is

$$L_V(\omega) = \mathbb{E}\left[\left(V_\omega(x) - (V_{\omega_k}(x) + A(x,u))\right)^2\right]. \tag{17}$$

The loss function for the policy network is

$$L_\pi(\theta) = -\mathbb{E}\left[\min\left\{\frac{\pi_\theta(u|x)}{\pi_{\theta_k}(u|x)}\bar{A}(x,u), \text{clip}\left(\frac{\pi_\theta(u|x)}{\pi_{\theta_k}(u|x)}, 1-\xi, 1+\xi\right)\bar{A}(x,u)\right\}\right], \tag{18}$$

where $\xi > 0$ is a constant for clipping the IS ratio. In the policy loss function, we use all state samples to approximate the advantage, which essentially replaces $d_0^\pi$ with $d^\pi$ in the objective function of Problem (11) for higher sample efficiency. The pseudocode of FPO is in Appendix B.2.

## 5 EXPERIMENTS

We aim to answer the following questions through our experiments:

**Q1** How does FPO perform in terms of safety and return compared to existing algorithms?

**Q2** Does FPO maintain monotonic expansion of the feasible region throughout training?

**Q3** What specific behaviors does FPO's policy learn to achieve both safety and high performance?

### 5.1 EXPERIMENT SETUPS

**Environments**  Our experiments cover 14 environments in the Safety-Gymnasium benchmark (Ji et al., 2023a), including navigation and locomotion. The navigation environments include two robots, i.e., Point and Car, and four tasks, i.e., Goal, Push, Button, and Circle, with all difficulty levels set as 1 and constraints set as default. The locomotion environments include six classic robots from Gymnasium's MuJoCo environments, i.e., HalfCheetah, Hopper, Swimmer, Walker2d, Ant, and Humanoid, with maximum velocity constraints.

**Baselines**  We compare FPO with a wide variety of mainstream safe RL algorithms implemented in the Omnisafe toolbox (Ji et al., 2023b), including iterative unconstrained RL methods RCPO (Tessler et al., 2018), PPO-Lag (Ray et al., 2019), and TRPO-PID (Stooke et al., 2020), and constrained policy optimization methods CPO (Achiam et al., 2017), PCPO (Yang et al., 2020), FOCOPS (Zhang et al., 2020), and P3O (Zhang et al., 2022). Hyperparameters for all algorithms are detailed in Appendix C.1. We use the default hyperparameters in Omnisafe for all baselines except that we set the cost limit to zero for all algorithms. Other hyperparameters have been tuned for good performance as stated by Ji et al. (2023b).

### 5.2 EXPERIMENT RESULTS

**Cost-return evaluation**  In safe RL, we evaluate algorithms by two metrics: (1) episode cost, representing the average number of constraint-violating steps per episode, and (2) episode return, representing the average cumulative rewards per episode. To perform a comprehensive evaluation, we place the scores of all algorithms in a cost-return plot in Figure 1. The scores are first normalized by those of PPO and then averaged on all 14 environments. The results demonstrate FPO's excellent performance in balancing safety and return: it reduces violation to 2% of PPO's level while maintaining 70% of its return. In contrast, other algorithms exhibit less favorable trade-offs. CPO and PCPO significantly sacrifice return due to their strict requirements on constraint satisfaction in every iteration. Lagrangian and penalty-based methods (PPO-Lag, RCPO, TRPO-PID, and

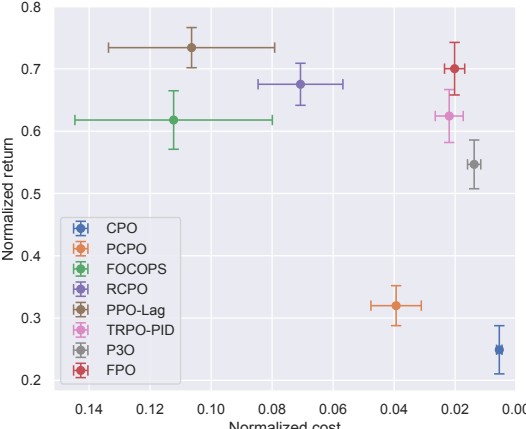

Figure 1: Normalized cost-return plot. The error bars represent 95% confidence intervals.

P3O) explicitly trade off cost and return by adjusting penalty coefficients, forming a Pareto front. Among these, TRPO-PID adaptively controls the Lagrange multiplier to achieve a more balanced performance, though it remains inferior to FPO in both safety and return. These results answer **Q1**.

**Training curves** Figure 2 shows the training curves of all algorithms across eight environments. Training curves on all 14 environments, along with final cost and return scores, are provided in Appendix C.2. FPO ideally balances cost and return in all environments. Notably, FPO is the only algorithm that finds a high-return and safe policy in SwimmerVelocity, while all other algorithms fails to solve this task. Constrained optimization methods like CPO and PCPO are overly conservative in most environments. Lagrangian-based methods like RCPO and PPO-Lag exhibit severe oscillations during training, resulting in inferior final performance. These results provide further empirical evidence to answer **Q1**.

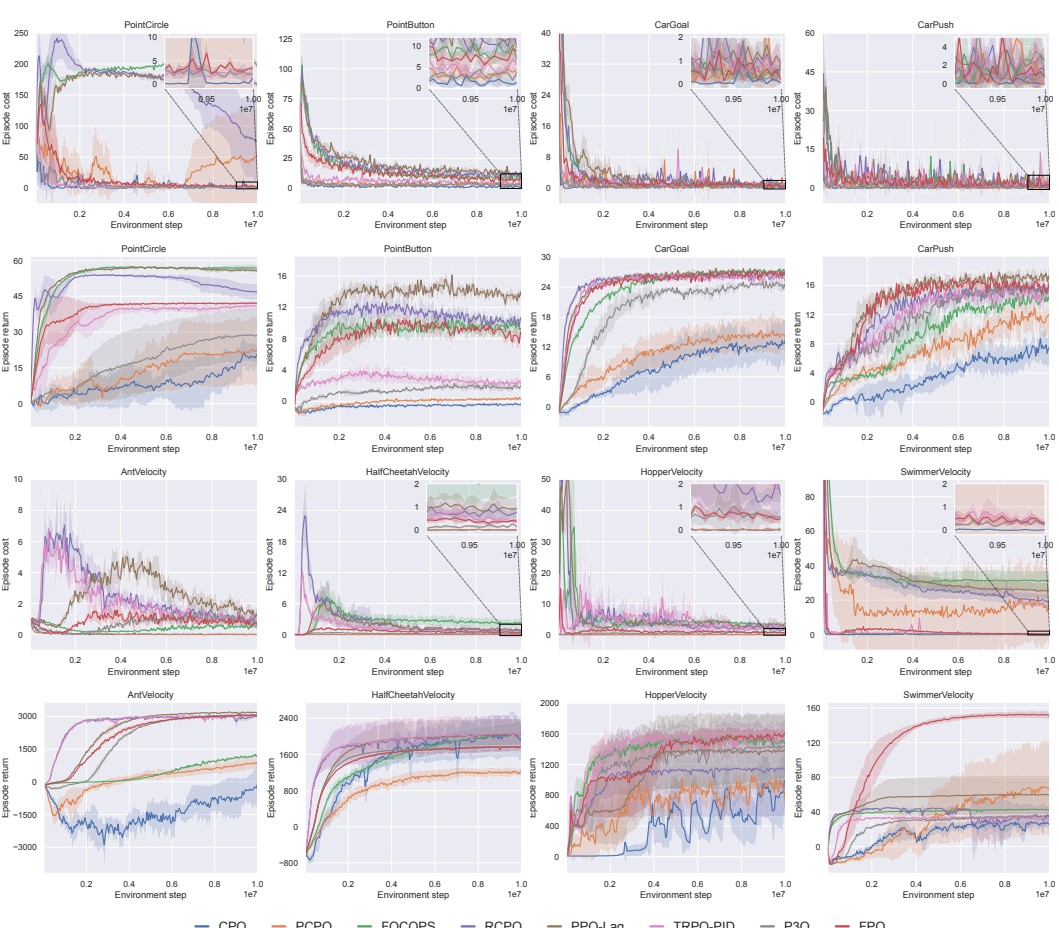

Figure 2: Training curves on eight environments in Safety-Gymnasium benchmark. The shaded areas represent 95% confidence intervals over 5 seeds.

**Feasible region visualization** We visualize the feasible regions learned by FPO during training in Figure 3 to check whether they are monotonically expanding as required by the constraint of our algorithm. While the training lasts 500 epochs, we find that the feasible regions after 100 epochs remain almost the same. The red circles in the figure are where the hazards are located. By epoch 5, FPO demonstrates preliminary capability to identify unsafe areas, but no state is identified as feasible. With continued learning, the feasible region emerges and gradually expands. By epoch 50, FPO already achieves complete distinguishability between feasible and infeasible regions. These results demonstrate that the monotonic expansion constraint of the feasible region is satisfied throughout

training, answering **Q2**. By quickly acquiring representations of the feasible region, FPO effectively focuses exploration within safe boundaries while optimizing returns.

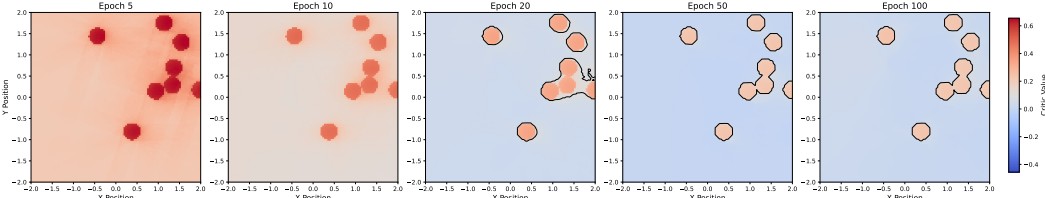

Figure 3: Visualization of the feasible regions during training in PointGoal. The colors represent CDF values computed by placing the agent on every point of a grid covering the space. The contours of the 0.1-level sets are marked in black.

**Trajectory visualization** We inspect the policy behavior in PointCircle and SwimmerVelocity by visualizing their trajectories. Figure 4 shows trajectories in PointCircle, where FPO follows circular motion that strictly stays within the constraint boundaries while PPO-Lag moves out of it. This violation occurs because learning a safe behavior in this task requires a quite large Lagrange multiplier, which PPO-Lag fails to reach within a limited training. FPO avoids this problem by directly constraining the policy inside the feasible region.

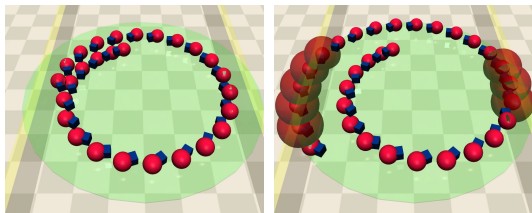

Figure 4: Trajectories of FPO (left) and PPO-Lag (right) in PointCircle.

Figure 5 shows trajectories in SwimmerVelocity, where FPO manages to move forward within speed limits while PPO-Lag is trapped in a local optimum. Specifically, PPO-Lag learns a policy that "climbs over" the constraint-violating pose as quickly as possible to reduce cumulative costs, before getting stuck in a safe pose with almost no rewards. FPO escapes this local optimum by avoiding any constraint violation in the first place. This owes to CDF, which treats all infeasible states equally, regardless of their future cumulative costs. These results answer **Q3**.

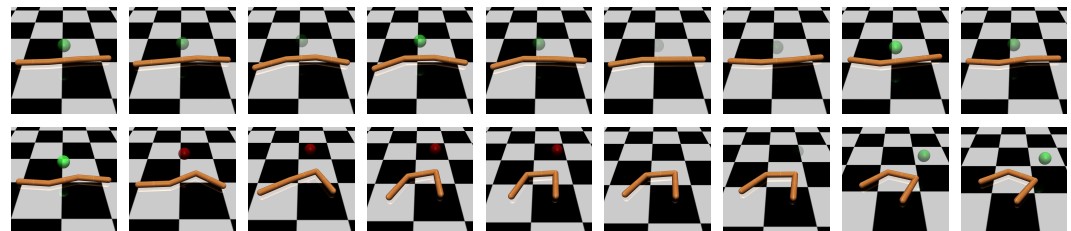

Figure 5: Trajectories of FPO (top) and PPO-Lag (bottom) in SwimmerVelocity.

## 6 CONCLUSION

This paper points out that the conventional practice of enforcing the original constraint in each iteration in safe RL is unnecessarily conservative. Instead, each update only needs to find an expanded feasible region and an improved value function. We propose an algorithm called FPO that achieves both objectives by simultaneously maximizing the value function inside the feasible region and minimizing the feasibility function outside it. We prove that these two optimization problems have a shared optimal solution, supported by a tight bound we derive on the CDF, which extends the result from CPO. Extensive experiments on Safety-Gymnasium show that FPO strikes a favorable balance between safety and return compared with state-of-the-art safe RL algorithms.

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

## A  PROOFS

### A.1  PROOF OF SHARED SOLUTION THEOREM

**Theorem 1.** *There exists a policy $\pi_{k+1}$ that is the optimal solution to both Problem (7) and (8).*

*Proof.* Let $\pi_{\text{in}}$ and $\pi_{\text{out}}$ denote the optimal solutions to Problem (7) and (8), respectively. We construct the policy $\pi_{k+1}$ as follows:

$$\pi_{k+1}(\cdot|x) = \begin{cases} \pi_{\text{in}}(\cdot|x), & x \in \mathcal{R}^{\pi_{\text{in}}}(\mathcal{X}_{\text{init}} \cap \mathrm{X}^{\pi_k}), \\ \pi_{\text{out}}(\cdot|x), & \text{otherwise,} \end{cases} \tag{19}$$

where $\mathcal{R}^{\pi}(X) = \bigcup_{x \in X} \mathcal{R}^{\pi}(x)$ denotes the reachable set of $\pi$ from a set of states $X \subseteq \mathcal{X}$.

By construction, the trajectories of $\pi_{k+1}$ starting from $\mathcal{X}_{\text{init}} \cap \mathrm{X}^{\pi_k}$ coincide with those of $\pi_{\text{in}}$. Therefore,

$$\mathbb{E}_{x \sim d_{\text{init}}}[\mathbb{I}[F^{\pi_k}(x) \leq 0]F^{\pi_{k+1}}(x)] = \mathbb{E}_{x \sim d_{\text{init}}}[\mathbb{I}[F^{\pi_k}(x) \leq 0]F^{\pi_{\text{in}}}(x)] \leq 0,$$

which proves that $\pi_{k+1}$ satisfies the shared constraint of both problems.

Since $\pi_{\text{in}}$ is optimal for Problem (7), and $\pi_{k+1}$ achieves the same value function as $\pi_{\text{in}}$ for all $x \in \mathcal{X}_{\text{init}} \cap \mathrm{X}^{\pi_k}$, it follows that

$$\mathbb{E}_{x \sim d_{\text{init}}}[\mathbb{I}[F^{\pi_k}(x) \leq 0]V^{\pi_{k+1}}(x)] = \mathbb{E}_{x \sim d_{\text{init}}}[\mathbb{I}[F^{\pi_k}(x) \leq 0]V^{\pi_{\text{in}}}(x)].$$

Thus, $\pi_{k+1}$ is also optimal for Problem (7).

For any $x \in \mathcal{X}_{\text{init}} \setminus \mathrm{X}^{\pi_k}$, we analyze two cases: (1) No future state enters $\mathcal{R}^{\pi_{\text{in}}}(\mathcal{X}_{\text{init}} \cap \mathrm{X}^{\pi_k})$. In this case, $\pi_{k+1} = \pi_{\text{out}}$ for all future states, thus $F^{\pi_{k+1}}(x) = F^{\pi_{\text{out}}}(x)$. (2) There exists a future state that enters $\mathcal{R}^{\pi_{\text{in}}}(\mathcal{X}_{\text{init}} \cap \mathrm{X}^{\pi_k})$ in finite time. In this case, $\pi_{k+1}$ switches to $\pi_{\text{in}}$ once entered, ensuring no future constraint violation. This, $F^{\pi_{k+1}}(x) \leq F^{\pi_{\text{out}}}(x)$. Combining these two cases, we have

$$\forall x \in \mathcal{X}_{\text{init}} \setminus \mathrm{X}^{\pi_k}, F^{\pi_{k+1}}(x) \leq F^{\pi_{\text{out}}}(x),$$

which implies

$$\mathbb{E}_{x \sim d_{\text{init}}}[\mathbb{I}[F^{\pi_k}(x) > 0]F^{\pi_{k+1}}(x)] \leq \mathbb{E}_{x \sim d_{\text{init}}}[\mathbb{I}[F^{\pi_k}(x) > 0]F^{\pi_{\text{out}}}(x)].$$

Since $\pi_{\text{out}}$ is optimal for Problem (8), $\pi_{k+1}$ is also optimal. Therefore, we conclude that $\pi_{k+1}$ is the optimal solution to both Problem (7) and (8). $\square$

### A.2  PROOF OF CDF BOUNDS

**Lemma 1.** *For any policies $\tilde{\pi}$ and $\pi$, and for any state $x \in \mathcal{X}$,*

$$F^{\tilde{\pi}}(x) - F^{\pi}(x) = \mathbb{E}_{\tau \sim \tilde{\pi}}\left[\sum_{t=0}^{\infty} \gamma^t \prod_{s=0}^{t-1}(1 - c_s)A_F^{\pi}(x_t, u_t)\Big|x_0 = x\right].$$

*Proof.* By definition of $F^{\pi}$, we have

$$F^{\pi}(x) = \mathbb{E}_{\tau \sim \pi}[c_0 + (1 - c_0)\gamma(c_1 + (1 - c_1)\gamma(\dots))|x_0 = x]$$

$$= \mathbb{E}_{\tau \sim \pi}[c_0 + \gamma(1 - c_0)c_1 + \gamma^2(1 - c_0)(1 - c_1)c_2 + \dots|x_0 = x]$$

$$= \mathbb{E}_{\tau \sim \pi}\left[\sum_{t=0}^{\infty} \gamma^t \prod_{s=0}^{t-1}(1 - c_s)c_t\Big|x_0 = x\right].$$

Thus,

$$F^{\tilde{\pi}}(x) - F^{\pi}(x) = \mathbb{E}_{\tau \sim \tilde{\pi}}\left[\sum_{t=0}^{\infty} \gamma^t \prod_{s=0}^{t-1}(1 - c_s)c_t\Big|x_0 = x\right] - F^{\pi}(x)$$

$$= \mathbb{E}_{\tau \sim \tilde{\pi}}\left[\sum_{t=0}^{\infty} \gamma^t \prod_{s=0}^{t-1}(1 - c_s)(c_t + (1 - c_t)\gamma F^{\pi}(x_{t+1}) - F^{\pi}(x_t))\Big|x_0 = x\right]$$

$$= \mathbb{E}_{\tau \sim \tilde{\pi}}\left[\sum_{t=0}^{\infty} \gamma^t \prod_{s=0}^{t-1}(1 - c_s)A_F^{\pi}(x_t, u_t)\Big|x_0 = x\right].$$

$\square$

**Definition 5** (Constraint-absorbing counterpart). *Let $M$ be an MDP with transition probability $P$. The constraint-absorbing counterpart of $M$, denoted $\tilde{M}$, is an MDP with all elements equal those of $M$ except the transition probability, which is defined as:*

$$
\tilde{P}(x'|x,u) = \begin{cases} P(x'|x,u), & c(x) = 0, \\ 1, & c(x) = 1 \text{ and } x' = x, \\ 0, & c(x) = 1 \text{ and } x' \neq x, \end{cases}
$$

*We also call such $\tilde{M}$ a constraint-absorbing MDP.*

**Lemma 2.** *In a constraint-absorbing MDP $\tilde{M}$, for any policies $\tilde{\pi}$ and $\pi$, and any state $x \in \mathcal{X}$,*

$$
\tilde{F}^{\tilde{\pi}}(x) - \tilde{F}^{\pi}(x) = \mathbb{E}_{\tau \sim (\tilde{\pi}, \tilde{P})} \left[ \sum_{t=0}^{\infty} \gamma^t \tilde{A}_F^{\pi}(x_t, u_t) \Big| x_0 = x \right].
$$

*Proof.* According to Lemma 1, we have

$$
\tilde{F}^{\tilde{\pi}}(x) - \tilde{F}^{\pi}(x) = \mathbb{E}_{\tau \sim (\tilde{\pi}, \tilde{P})} \left[ \sum_{t=0}^{\infty} \gamma^t \prod_{s=0}^{t-1}(1 - c_s) \tilde{A}_F^{\pi}(x_t, u_t) \Big| x_0 = x \right].
$$

We split the summation into two parts:

$$
\tilde{F}^{\tilde{\pi}}(x) - \tilde{F}^{\pi}(x) = \mathbb{E}_{\tau \sim (\tilde{\pi}, \tilde{P})} \Bigg[ \sum_{t=0}^{N(\tau)} \gamma^t \prod_{s=0}^{t-1}(1 - c_s) \tilde{A}_F^{\pi}(x_t, u_t)
$$
$$
+ \sum_{t=N(\tau)+1}^{\infty} \gamma^t \prod_{s=0}^{t-1}(1 - c_s) \tilde{A}_F^{\pi}(x_t, u_t) \Big| x_0 = x \Bigg].
$$

For any trajectory $\tau \sim (\tilde{\pi}, \tilde{P})$, for all $t \leq N(\tau)$, we have $c_{t-1} = 0$. Thus,

$$
\sum_{t=0}^{N(\tau)} \gamma^t \prod_{s=0}^{t-1}(1 - c_s) \tilde{A}_F^{\pi}(x_t, u_t) = \sum_{t=0}^{N(\tau)} \gamma^t \tilde{A}_F^{\pi}(x_t, u_t).
$$

For all $t > N(\tau)$, since $c_{N(\tau)} = 1$, we have

$$
\sum_{t=N(\tau)+1}^{\infty} \gamma^t \prod_{s=0}^{t-1}(1 - c_s) \tilde{A}_F^{\pi}(x_t, u_t) = 0.
$$

By definition of $\tilde{P}$, for all $t > N(\tau)$, we have $x_t = x_{N(\tau)}$, $\tilde{F}^{\pi}(x_t) = c_t = 1$, and it follows that

$$
\tilde{A}_F^{\pi}(x_t, u_t) = \mathbb{E}_{x_{t+1} \sim \tilde{P}(\cdot|x_t, u_t)}[c_t + (1 - c_t)\gamma \tilde{F}^{\pi}(x_{t+1}) - \tilde{F}^{\pi}(x_t)] = 0.
$$

Thus, we can equivalently write the second half of the summation as follows:

$$
\sum_{t=N(\tau)+1}^{\infty} \gamma^t \prod_{s=0}^{t-1}(1 - c_s) \tilde{A}_F^{\pi}(x_t, u_t) = \sum_{t=N(\tau)+1}^{\infty} \gamma^t \tilde{A}_F^{\pi}(x_t, u_t).
$$

Therefore, we conclude that

$$
\tilde{F}^{\tilde{\pi}}(x) - \tilde{F}^{\pi}(x) = \mathbb{E}_{\tau \sim (\tilde{\pi}, \tilde{P})} \left[ \sum_{t=0}^{\infty} \gamma^t \tilde{A}_F^{\pi}(x_t, u_t) \Big| x_0 = x \right].
$$

$\square$

**Lemma 3** (CDF equivalence). *Let $F^{\pi}$ be the CDF in an MDP $M$, and $\tilde{F}^{\pi}$ be the CDF in $\tilde{M}$. For any policy $\pi$ and state $x \in \mathcal{X}$, we have*

$$
\tilde{F}^{\pi}(x) = F^{\pi}(x).
$$

*Proof.* Consider a trajectory where the first constraint violation happens at time step $t$, and we denote it as $\tau_t$, i.e., $\tau_t = \{x_0, u_0, x_1, u_1, \dots\}$, where $c(x_t) = 1$ and $c(x_s) = 0, \forall s < t$. We split $\tau_t$ into two parts:

$$\tau_{\leq t} = \{x_0, u_0, x_1, u_1, \dots, x_t\} \text{ and } \tau_{>t} = \{u_t, x_{t+1}, u_{t+1}, \dots\}.$$

The probability of $\tau_t$ under the original MDP $M$ can be decomposed as follows:

$$p(\tau_t) = p(\tau_{\leq t})p(\tau_{>t}|\tau_{\leq t}),$$

where

$$p(\tau_{\leq t}) = \mathbb{I}[x_0 = x] \prod_{s=0}^{t-1} \pi(u_s|x_s)P(x_{s+1}|x_s, u_s),$$

$$p(\tau_{>t}|\tau_{\leq t}) = \prod_{s=t}^{\infty} \pi(u_s|x_s)P(x_{s+1}|x_s, u_s).$$

Using the decomposed probability, the CDF can be expressed as:

$$F^\pi(x) = \sum_{\tau} p(\tau)\gamma^{N(\tau)}$$

$$= \sum_{t=0}^{\infty} \sum_{\tau_t} p(\tau_t)\gamma^t$$

$$= \sum_{t=0}^{\infty} \sum_{\tau_{\leq t}, \tau_{>t}} p(\tau_{\leq t})p(\tau_{>t}|\tau_{\leq t})\gamma^t$$

$$= \sum_{t=0}^{\infty} \sum_{\tau_{\leq t}} p(\tau_{\leq t}) \underbrace{\sum_{\tau_{>t}} p(\tau_{>t}|\tau_{\leq t})}_{=1} \gamma^t$$

$$= \sum_{t=0}^{\infty} \sum_{\tau_{\leq t}} p(\tau_{\leq t})\gamma^t.$$

Similarly, the CDF in $\tilde{M}$ can be expressed as

$$\tilde{F}^\pi(x) = \sum_{t=0}^{\infty} \sum_{\tau_{\leq t}} \tilde{p}(\tau_{\leq t})\gamma^t,$$

where

$$\tilde{p}(\tau_{\leq t}) = \mathbb{I}[x_0 = x] \prod_{s=0}^{t-1} \pi(u_s|x_s)\tilde{P}(x_{s+1}|x_s, u_s).$$

Since the transition probability $\tilde{P}$ is identical to $P$ up to the first constraint violation, we have $\tilde{p}(\tau_{\leq t}) = p(\tau_{\leq t})$, and thus $\tilde{F}^\pi(x) = F^\pi(x)$. $\qquad\square$

**Lemma 4** (Feasibility advantage equivalence). *Let $A_F^\pi$ be the feasibility advantage in an MDP $M$, and $\tilde{A}_F^\pi$ be the feasibility advantage in $\tilde{M}$. For any policy $\pi$, state $x \in \mathcal{X}$, and action $u \in \mathcal{U}$, we have*

$$\tilde{A}_F^\pi(x, u) = A_F^\pi(x, u).$$

*Proof.* By definition of the feasibility advantage,

$$\tilde{A}_F^\pi(x, u) = \mathbb{E}_{x' \sim \tilde{P}(\cdot|x,u)}[c(x) + (1 - c(x))\gamma\tilde{F}^\pi(x') - \tilde{F}^\pi(x)].$$

By Lemma 3, we can replace $\tilde{F}^\pi$ with $F^\pi$:

$$\tilde{A}_F^\pi(x, u) = \mathbb{E}_{x' \sim \tilde{P}(\cdot|x,u)}[c(x) + (1 - c(x))\gamma F^\pi(x') - F^\pi(x)].$$

Now, the only difference between $\tilde{A}_F^\pi(x, u)$ and $A_F^\pi(x, u)$ lies in the transition probability. We analyze two cases: whether state $x$ violates the constraint or not. If $c(x) = 0$, we have $\tilde{P}(\cdot|x, u) = P(\cdot|x, u)$. In this case, $\tilde{A}_F^\pi(x, u) = A_F^\pi(x, u)$. If $c(x) = 1$, we have $F^\pi(x) = 1$. In this case,

$$c(x) + (1 - c(x))\gamma F^\pi(x') - F^\pi(x) = 0,$$

and thus $\tilde{A}_F^\pi(x, u) = A_F^\pi(x, u) = 0$. Therefore, $\tilde{A}_F^\pi(x, u) = A_F^\pi(x, u)$ holds for all $x \in \mathcal{X}$. $\qquad\square$

**Lemma 5** (Prefix state distribution equivalence). *Let $d_0^\pi$ be the prefix state distribution in an MDP $M$, and $\tilde{d}_0^\pi$ be the prefix state distribution in $\tilde{M}$. For any policy $\pi$, initial state $x \in \mathcal{X}$, and future state $x' \in \mathcal{X}$, we have*

$$\tilde{d}_0^\pi(x'|x) = d_0^\pi(x'|x).$$

*Proof.* Expand the probability in each term of the summation,

$$P(x_t = x', \max_{s<t} c_s = 0|x, \pi) = \sum_{\substack{x_1, x_2, \ldots, x_{t-1} \in \mathrm{X_{cstr}} \\ u_0, u_1, \ldots, u_{t-1} \in \mathcal{U}}} \pi(u_0|x)P(x_1|x, u_0)\pi(u_1|x_1) \cdots P(x'|x_{t-1}, u_{t-1}).$$

Since $c_s = 0, \forall s < t$, by definition of $\tilde{P}$, we have

$$\tilde{P}(x_{s+1}|x_s, u_s) = P(x_{s+1}|x_s, u_s), \forall s < t.$$

Thus, it follows that

$$\tilde{P}(x_t = x', \max_{s<t} c_s = 0|x, \pi) = P(x_t = x', \max_{s<t} c_s = 0|x, \pi),$$

which implies that $\tilde{d}_0^\pi(x'|x) = d_0^\pi(x'|x)$. $\qquad\square$

**Lemma 6.** *For any policies $\tilde{\pi}$ and $\pi$, and state $x \in \mathcal{X}$,*

$$F^{\tilde{\pi}}(x) - F^\pi(x) = \frac{1}{1 - \gamma}\mathbb{E}_{x' \sim d_0^{\tilde{\pi}}(\cdot|x), u' \sim \tilde{\pi}(\cdot|x')}[A_F^\pi(x', u')].$$

*Proof.* By Lemma 2, we have

$$\tilde{F}^{\tilde{\pi}}(x) - \tilde{F}^\pi(x) = \mathbb{E}_{\tau \sim (\tilde{\pi}, \tilde{P})}\left[\sum_{t=0}^\infty \gamma^t \tilde{A}_F^\pi(x_t, u_t)\Big|x_0 = x\right]$$

$$= \sum_{t=0}^\infty \sum_{x'} \tilde{P}(x_t = x'|x, \tilde{\pi}) \sum_{u'} \tilde{\pi}(u'|x')\gamma^t \tilde{A}_F^\pi(x', u').$$

For any $t \geq 0$, if $\max_{s<t} c_s > 0$, the state will be fixed at the constraint-violating one in the constraint absorbing MDP. Thus, only those $x'$ that violate the constraint yield $\tilde{P}(x_t = x'|x, \tilde{\pi}) > 0$. For these $x'$, we have $\tilde{A}_F^\pi(x', u') = 0$. Therefore, we only need to consider the terms with $\max_{s<t} c_s = 0$ in the summation, i.e.,

$$\tilde{F}^{\tilde{\pi}}(x) - \tilde{F}^\pi(x) = \sum_{t=0}^\infty \sum_{x'} \tilde{P}(x_t = x'|x, \tilde{\pi}, \max_{s<t} c_s = 0) \sum_{u'} \tilde{\pi}(u'|x')\gamma^t \tilde{A}_F^\pi(x', u')$$

$$= \sum_{x'} \sum_{t=0}^\infty \gamma^t \tilde{P}(x_t = x'|x, \tilde{\pi}, \max_{s<t} c_s = 0) \sum_{u'} \tilde{\pi}(u'|x')\tilde{A}_F^\pi(x', u')$$

$$= \sum_{x'} \frac{1}{1 - \gamma}\tilde{d}_0^{\tilde{\pi}}(x'|x) \sum_{u'} \tilde{\pi}(u'|x')\tilde{A}_F^\pi(x', u')$$

$$= \frac{1}{1 - \gamma}\mathbb{E}_{x' \sim \tilde{d}_0^{\tilde{\pi}}(\cdot|x), u' \sim \tilde{\pi}(\cdot|x')}[\tilde{A}_F^\pi(x', u')].$$

Substitute in the result from Lemma 3, 4, and 5, we have

$$F^{\tilde{\pi}}(x) - F^\pi(x) = \frac{1}{1 - \gamma}\mathbb{E}_{x' \sim d_0^{\tilde{\pi}}(\cdot|x), u' \sim \tilde{\pi}(\cdot|x')}[A_F^\pi(x', u')].$$

$\qquad\square$

**Lemma 7.** *For any policies $\tilde{\pi}$ and $\pi$, and any state $x \in \mathcal{X}$, define*

$$L_{\tilde{\pi}}^{\pi}(x) = \mathbb{E}_{x' \sim d_0^{\pi}(\cdot|x), u' \sim \pi(\cdot|x')} \left[ \frac{\tilde{\pi}(u'|x')}{\pi(u'|x')} A_F^{\pi}(x', u') \right],$$

*and $\epsilon_F^{\tilde{\pi}} = \max_{x'} |\mathbb{E}_{u' \sim \tilde{\pi}(\cdot|x')}[A_F^{\pi}(x', u')]|$. The following bounds hold:*

$$F^{\tilde{\pi}}(x) - F^{\pi}(x) \geq \frac{1}{1 - \gamma} \left( L_{\tilde{\pi}}^{\pi}(x) - 2\epsilon_F^{\tilde{\pi}} D_{TV}(d_0^{\tilde{\pi}}(\cdot|x)\|d_0^{\pi}(\cdot|x)) \right),$$

$$F^{\tilde{\pi}}(x) - F^{\pi}(x) \leq \frac{1}{1 - \gamma} \left( L_{\tilde{\pi}}^{\pi}(x) + 2\epsilon_F^{\tilde{\pi}} D_{TV}(d_0^{\tilde{\pi}}(\cdot|x)\|d_0^{\pi}(\cdot|x)) \right),$$

*where $D_{TV}$ is the total variational divergence. Furthermore, the bounds are tight (when $\tilde{\pi} = \pi$, the LHS and RHS are identically zero).*

*Proof.* This proof is largely borrowed from Lemma 2 in CPO (Achiam et al., 2017).

Let $\bar{A}_F^{\pi} \in \mathbb{R}^{|\mathcal{X}|}$ denote the vector of components $\bar{A}_F^{\pi}(x') = \mathbb{E}_{u' \sim \tilde{\pi}(\cdot|x')}[A_F^{\pi}(x', u')]$. With an abuse of notation, we view $d_0^{\pi}(\cdot|x)$ as a vector in $\mathbb{R}^{|\mathcal{X}|}$ when necessary. Beginning with the result in Lemma 6, we have

$$(1 - \gamma)(F^{\tilde{\pi}}(x) - F^{\pi}(x)) = \mathbb{E}_{x' \sim d_0^{\tilde{\pi}}(\cdot|x), u' \sim \tilde{\pi}(\cdot|x')}[A_F^{\pi}(x', u')]$$
$$= \langle d_0^{\tilde{\pi}}(\cdot|x), \bar{A}_F^{\pi} \rangle$$
$$= \langle d_0^{\pi}(\cdot|x), \bar{A}_F^{\pi} \rangle + \langle d_0^{\tilde{\pi}}(\cdot|x) - d_0^{\pi}(\cdot|x), \bar{A}_F^{\pi} \rangle.$$

This term can be bounded by Holder's inequality: for any $p, q \in [1, \infty]$ such that $1/p + 1/q = 1$, we have

$$(1 - \gamma)(F^{\tilde{\pi}}(x) - F^{\pi}(x)) \geq \langle d_0^{\pi}(\cdot|x), \bar{A}_F^{\pi} \rangle - \left\| d_0^{\tilde{\pi}}(\cdot|x) - d_0^{\pi}(\cdot|x) \right\|_p \left\| \bar{A}_F^{\pi} \right\|_q,$$

$$(1 - \gamma)(F^{\tilde{\pi}}(x) - F^{\pi}(x)) \leq \langle d_0^{\pi}(\cdot|x), \bar{A}_F^{\pi} \rangle + \left\| d_0^{\tilde{\pi}}(\cdot|x) - d_0^{\pi}(\cdot|x) \right\|_p \left\| \bar{A}_F^{\pi} \right\|_q.$$

Choose $p = 1, q = \infty$, we have $\|d_0^{\tilde{\pi}}(\cdot|x) - d_0^{\pi}(\cdot|x)\|_1 = 2D_{TV}(d_0^{\tilde{\pi}}(\cdot|x)\|d_0^{\pi}(\cdot|x))$ and $\|\bar{A}_F^{\pi}\|_{\infty} = \epsilon_F^{\tilde{\pi}}$. Observe that by importance sampling,

$$\langle d_0^{\pi}(\cdot|x), \bar{A}_F^{\pi} \rangle = \mathbb{E}_{x' \sim d_0^{\pi}(\cdot|x), u' \sim \tilde{\pi}(\cdot|x')}[A_F^{\pi}(x', u')]$$
$$= \mathbb{E}_{x' \sim d_0^{\pi}(\cdot|x), u' \sim \pi(\cdot|x')} \left[ \frac{\tilde{\pi}(u'|x')}{\pi(u'|x')} A_F^{\pi}(x', u') \right]$$
$$= L_{\tilde{\pi}}^{\pi}(x).$$

After rearranging terms, the bounds are obtained. $\qquad\square$

**Lemma 8.** *For any policies $\tilde{\pi}$ and $\pi$, and state $x \in \mathcal{X}$,*

$$\left\| d_0^{\tilde{\pi}}(\cdot|x) - d_0^{\pi}(\cdot|x) \right\|_1 \leq \frac{2\gamma}{1 - \gamma} \mathbb{E}_{x' \sim d_0^{\pi}(\cdot|x)} \left[ D_{TV}(\tilde{\pi}\|\pi)[x'] \right],$$

*where $D_{TV}(\tilde{\pi}\|\pi)[x'] = (1/2) \sum_u |\tilde{\pi}(u|x') - \pi(u|x')|$.*

*Proof.* We prove that the result holds for the prefix state distribution in a constraint-absorbing MDP, i.e., $\tilde{d}_0^{\tilde{\pi}}$ and $\tilde{d}_0^{\pi}$. Since $\tilde{d}_0^{\pi} = d_0^{\pi}$ for any $\pi$, the result to prove directly follows.

Let $\tilde{P}^{\pi}(x'|x) = \sum_u \tilde{P}(x'|x, u)\pi(u|x)$. We view $\tilde{P}^{\pi}$ as a matrix in $\mathbb{R}^{|\mathcal{X}| \times |\mathcal{X}|}$, where the element on the $i$th row and $j$th column, $\tilde{P}_{ij}^{\pi}$, denotes the transition probability from the $j$th state to the $i$th state. We rearrange the order of the states in $\tilde{P}^{\pi}$ so that all constraint-violating states are located on the last rows and columns:

$$\tilde{P}^{\pi} = \begin{bmatrix} \tilde{P}_s^{\pi} & O \\ \tilde{P}_v^{\pi} & I \end{bmatrix},$$

where $\tilde{P}_s^{\pi}$ denotes the transition probability between constraint-satisfying states, $\tilde{P}_v^{\pi}$ denotes the transition probability from constraint-satisfying states to constraint-violating states, $O$ denotes the

zero matrix, and $I$ denotes the identity matrix, which implies that a constraint-violating state will no longer transfer to other states. We construct another matrix by setting the identity matrix in $\tilde{P}^\pi$ to zero:

$$\tilde{P}_0^\pi = \begin{bmatrix} \tilde{P}_s^\pi & O \\ \tilde{P}_v^\pi & O \end{bmatrix}.$$

By definition of the prefix state distribution,

$$\tilde{d}_0^\pi(\cdot|x) = (1-\gamma)\sum_{t=0}^\infty \left(\gamma\tilde{P}_0^\pi\right)^t e_x = (1-\gamma)\left(I - \gamma\tilde{P}_0^\pi\right)^{-1} e_x,$$

where $e_x$ is a one-hot vector where the element at the position of state $x$ is one, and all other elements are zero, which implies that the initial state is fixed at $x$.

Define matrices $G = (I - \gamma\tilde{P}_0^\pi)^{-1}$, $\tilde{G} = (I - \gamma\tilde{P}_0^{\tilde{\pi}})^{-1}$, and $\Delta = \tilde{P}_0^{\tilde{\pi}} - \tilde{P}_0^\pi$. Then,

$$G^{-1} - \tilde{G}^{-1} = \left(I - \gamma\tilde{P}_0^\pi\right) - \left(I - \gamma\tilde{P}_0^{\tilde{\pi}}\right) = \gamma\Delta.$$

Left-multiplying by $G$ and right-multiplying by $\tilde{G}$, we obtain

$$\tilde{G} - G = \gamma\tilde{G}\Delta G.$$

Thus,

$$\begin{aligned}
\tilde{d}_0^{\tilde{\pi}}(\cdot|x) - \tilde{d}_0^\pi(\cdot|x) &= (1-\gamma)\left(\tilde{G} - G\right)e_x \\
&= \gamma(1-\gamma)\tilde{G}\Delta G e_x \\
&= \gamma\tilde{G}\Delta\tilde{d}_0^\pi(\cdot|x).
\end{aligned}$$

Taking the L1 norm on both sides, we obtain

$$\left\|\tilde{d}_0^{\tilde{\pi}}(\cdot|x) - \tilde{d}_0^\pi(\cdot|x)\right\|_1 = \gamma\left\|\tilde{G}\Delta\tilde{d}_0^\pi(\cdot|x)\right\|_1 \le \gamma\left\|\tilde{G}\right\|_1\left\|\Delta\tilde{d}_0^\pi(\cdot|x)\right\|_1.$$

$\|\tilde{G}\|_1$ is bounded by

$$\left\|\tilde{G}\right\|_1 = \left\|\left(I - \gamma\tilde{P}_0^{\tilde{\pi}}\right)^{-1}\right\|_1 \le \sum_{t=0}^\infty \gamma^t\left\|\tilde{P}_0^{\tilde{\pi}}\right\|_1^t \le \sum_{t=0}^\infty \gamma^t = (1-\gamma)^{-1}.$$

$\|\Delta\tilde{d}_0^\pi(\cdot|x)\|_1$ is bounded by

$$\begin{aligned}
\left\|\Delta\tilde{d}_0^\pi(\cdot|x)\right\|_1 &= \sum_{x''}\left|\sum_{x'}\Delta(x''|x')\tilde{d}_0^\pi(x'|x)\right| \\
&\le \sum_{x',x''}|\Delta(x''|x')|\,\tilde{d}_0^\pi(x'|x) \\
&= \sum_{x',x''}\left|\sum_{u'}\tilde{P}(x''|x',u')(\tilde{\pi}(u'|x') - \pi(u'|x'))\right|\tilde{d}_0^\pi(x'|x) \\
&\le \sum_{x',u',x''}P(x''|x',u')\,|\tilde{\pi}(u'|x') - \pi(u'|x')|\,\tilde{d}_0^\pi(x'|x) \\
&= \sum_{x',u'}|\tilde{\pi}(u'|x') - \pi(u'|x')|\,\tilde{d}_0^\pi(x'|x) \\
&= 2\mathbb{E}_{x'\sim\tilde{d}_0^\pi(\cdot|x)}\left[D_{TV}(\tilde{\pi}\|\pi)[x']\right].
\end{aligned}$$

Therefore,

$$\left\|\tilde{d}_0^{\tilde{\pi}}(\cdot|x) - \tilde{d}_0^\pi(\cdot|x)\right\|_1 \le \frac{2\gamma}{1-\gamma}\mathbb{E}_{x'\sim\tilde{d}_0^\pi(\cdot|x)}\left[D_{TV}(\tilde{\pi}\|\pi)[x']\right].$$

$\square$

**Theorem 2.** *For any policies $\tilde{\pi}$ and $\pi$, and any state $x \in \mathcal{X}$, define*

$$A_F^\pi(x,u) = \mathbb{E}_{x'\sim P(\cdot|x,u)}[c(x) + (1-c(x))\gamma F^\pi(x') - F^\pi(x)],$$

*and $L_{\tilde{\pi}}^\pi(x) = \mathbb{E}_{x'\sim d_0^\pi(\cdot|x), u'\sim\tilde{\pi}(\cdot|x')}[A_F^\pi(x',u')]$, $\epsilon_F^{\tilde{\pi}} = \max_x |\mathbb{E}_{u\sim\tilde{\pi}(\cdot|x)}[A_F^\pi(x,u)]|$. Then,*

$$F^{\tilde{\pi}}(x) - F^\pi(x) \geq \frac{L_{\tilde{\pi}}^\pi(x)}{1-\gamma} - \frac{2\gamma\epsilon_F^{\tilde{\pi}}}{(1-\gamma)^2}\mathbb{E}_{x'\sim d_0^\pi(\cdot|x)}[D_{TV}(\tilde{\pi}\|\pi)[x']],$$

$$F^{\tilde{\pi}}(x) - F^\pi(x) \leq \frac{L_{\tilde{\pi}}^\pi(x)}{1-\gamma} + \frac{2\gamma\epsilon_F^{\tilde{\pi}}}{(1-\gamma)^2}\mathbb{E}_{x'\sim d_0^\pi(\cdot|x)}[D_{TV}(\tilde{\pi}\|\pi)[x']],$$

*where $D_{TV}(\tilde{\pi}\|\pi)[x'] = (1/2)\sum_u |\tilde{\pi}(u|x') - \pi(u|x')|$ is the total variational divergence between action distributions at $x'$. Furthermore, the bounds are tight (when $\tilde{\pi} = \pi$, the LHS and RHS are identically zero).*

*Proof.* Begin with the bounds from Lemma 7 and bound the divergence by Lemma 8. □

### A.3 PROOF OF PERFORMANCE BOUNDS

**Corollary 2.** *The optimal solution to Problem (11), denoted $\pi_{k+1}$, satisfies the following two properties:*

*1. Feasibility enhancement:*

$$\mathbb{E}_{x\sim d_{init}}\left[\mathbb{I}[F^{\pi_k}(x) \leq 0]F^{\pi_{k+1}}(x)\right] \leq \frac{\sqrt{\delta}\gamma\epsilon_F^{\pi_{k+1}}}{(1-\gamma)^2}, \tag{12a}$$

$$\mathbb{E}_{x\sim d_{init}}\left[\mathbb{I}[F^{\pi_k}(x) > 0]\left(F^{\pi_{k+1}}(x) - F^{\pi_k}(x)\right)\right] \leq \frac{\sqrt{\delta}\gamma\epsilon_F^{\pi_{k+1}}}{(1-\gamma)^2}, \tag{12b}$$

*where $\epsilon_F^{\pi_{k+1}} = \max_x |\mathbb{E}_{u\sim\pi_{k+1}(\cdot|x)}[A_F^\pi(x,u)]|$.*

*2. Value improvement:*

$$\mathbb{E}_{x\sim d_{init}}\left[\mathbb{I}[F^{\pi_k}(x) \leq 0]\left(V^{\pi_{k+1}}(x) - V^{\pi_k}(x)\right)\right] \geq -\frac{\sqrt{\delta}\gamma\epsilon^{\pi_{k+1}}}{(1-\gamma)^2}, \tag{13}$$

*where $\epsilon^{\pi_{k+1}} = \max_x |\mathbb{E}_{u\sim\pi_{k+1}(\cdot|x)}[A^\pi(x,u)]|$.*

*Proof.* Consider the following two problems:

$$\max_\pi \mathbb{E}_{x\sim d^{\pi_k}, u\sim\pi}\left[\mathbb{I}[F^{\pi_k}(x) \leq 0]A^{\pi_k}(x,u)\right]$$

$$\text{s.t. } \mathbb{E}_{x\sim d^{\pi_k}, u\sim\pi}\left[\mathbb{I}[F^{\pi_k}(x) \leq 0](F^{\pi_k}(x) + A_F^{\pi_k}(x,u)/(1-\gamma))_+\right] \leq 0$$

$$\mathbb{E}_{x\sim d^{\pi_k}}\left[\mathbb{I}[F^{\pi_k}(x) \leq 0]D_{KL}(\pi\|\pi_k)[x]\right] \leq \delta/2 \tag{20}$$

$$\mathbb{E}_{x\sim d^{\pi_k}}\left[\mathbb{I}[F^{\pi_k}(x) > 0]D_{KL}(\pi\|\pi_k)[x]\right] \leq \delta/2,$$

and

$$\min_\pi \mathbb{E}_{x\sim d_0^{\pi_k}, u\sim\pi}\left[\mathbb{I}[F^{\pi_k}(x) > 0]A_F^{\pi_k}(x,u)\right]$$

$$\text{s.t. } \mathbb{E}_{x\sim d^{\pi_k}, u\sim\pi}\left[\mathbb{I}[F^{\pi_k}(x) \leq 0](F^{\pi_k}(x) + A_F^{\pi_k}(x,u)/(1-\gamma))_+\right] \leq 0$$

$$\mathbb{E}_{x\sim d^{\pi_k}}\left[\mathbb{I}[F^{\pi_k}(x) \leq 0]D_{KL}(\pi\|\pi_k)[x]\right] \leq \delta/2 \tag{21}$$

$$\mathbb{E}_{x\sim d^{\pi_k}}\left[\mathbb{I}[F^{\pi_k}(x) > 0]D_{KL}(\pi\|\pi_k)[x]\right] \leq \delta/2,$$

We prove that they have the same optimal solution. Let $\pi_{\text{in}}$ and $\pi_{\text{out}}$ denote the optimal solutions to Problem (20) and (21), respectively. Construct the following policy:

$$\pi_{k+1}(\cdot|x) = \begin{cases} \pi_{\text{in}}(\cdot|x), & x \in \mathrm{X}^{\pi_k}, \\ \pi_{\text{out}}(\cdot|x), & \text{otherwise.} \end{cases}$$

We first prove that $\pi_{k+1}$ satisfies the constraints of Problem (20) and (21). For the first constraint, we have

$$\mathbb{E}_{x\sim d^{\pi_k}, u\sim\pi_{k+1}}\left[\mathbb{I}[F^{\pi_k}(x) \leq 0](F^{\pi_k}(x) + A_F^{\pi_k}(x,u)/(1-\gamma))_+\right]$$

$$= \mathbb{E}_{x\sim d^{\pi_k}, u\sim\pi_{\text{in}}}\left[\mathbb{I}[F^{\pi_k}(x) \leq 0](F^{\pi_k}(x) + A_F^{\pi_k}(x,u)/(1-\gamma))_+\right] \leq 0.$$

For the second and third constraints, we have

$$\mathbb{E}_{x \sim d^{\pi_k}}\left[\mathbb{I}[F^{\pi_k}(x) \le 0] D_{KL}(\pi_{k+1} \| \pi_k)[x]\right] = \mathbb{E}_{x \sim d^{\pi_k}}\left[\mathbb{I}[F^{\pi_k}(x) \le 0] D_{KL}(\pi_{\text{in}} \| \pi_k)[x]\right] \le \delta/2,$$
$$\mathbb{E}_{x \sim d^{\pi_k}}\left[\mathbb{I}[F^{\pi_k}(x) > 0] D_{KL}(\pi_{k+1} \| \pi_k)[x]\right] = \mathbb{E}_{x \sim d^{\pi_k}}\left[\mathbb{I}[F^{\pi_k}(x) > 0] D_{KL}(\pi_{\text{out}} \| \pi_k)[x]\right] \le \delta/2.$$

Thus, $\pi_{k+1}$ satisfies the constraints of both problems. For the objective function of Problem (20), we have

$$\mathbb{E}_{x \sim d^{\pi_k}, u \sim \pi_{k+1}}\left[\mathbb{I}[F^{\pi_k}(x) \le 0] A^{\pi_k}(x, u)\right] = \mathbb{E}_{x \sim d^{\pi_k}, u \sim \pi_{\text{in}}}\left[\mathbb{I}[F^{\pi_k}(x) \le 0] A^{\pi_k}(x, u)\right],$$

which proves that $\pi_{k+1}$ is the optimal solution to Problem (20). For the objective function of Problem (21), we have

$$\mathbb{E}_{x \sim d_0^{\pi_k}, u \sim \pi_{k+1}}\left[\mathbb{I}[F^{\pi_k}(x) > 0] A_F^{\pi_k}(x, u)\right] = \mathbb{E}_{x \sim d_0^{\pi_k}, u \sim \pi_{\text{out}}}\left[\mathbb{I}[F^{\pi_k}(x) > 0] A_F^{\pi_k}(x, u)\right],$$

which proves that $\pi_{k+1}$ is the optimal solution to Problem (21). Thus, $\pi_{k+1}$ is the optimal solution to both Problem (20) and (21). Since the original problem (11) is the summation of Problem (20) and (21), $\pi_{k+1}$ is also the optimal solution to Problem (11).

As the optimal solution to Problem (20) and (21), $\pi_{k+1}$ must be better than any other feasible solution to these two problems. Specifically, it must be better $\pi_k$. Since

$$\mathbb{E}_{u \sim \pi_k}\left[A^{\pi_k}(x, u)\right] = \mathbb{E}_{u \sim \pi_k}\left[A_F^{\pi_k}(x, u)\right] = 0, \forall x \in \mathcal{X},$$

we have

$$\mathbb{E}_{x \sim d^{\pi_k}, u \sim \pi_{k+1}}\left[\mathbb{I}[F^{\pi_k}(x) > 0] A^{\pi_k}(x, u)\right] \ge \mathbb{E}_{x \sim d^{\pi_k}, u \sim \pi_k}\left[\mathbb{I}[F^{\pi_k}(x) > 0] A^{\pi_k}(x, u)\right] = 0,$$
$$\mathbb{E}_{x \sim d_0^{\pi_k}, u \sim \pi_{k+1}}\left[\mathbb{I}[F^{\pi_k}(x) > 0] A_F^{\pi_k}(x, u)\right] \le \mathbb{E}_{x \sim d_0^{\pi_k}, u \sim \pi_k}\left[\mathbb{I}[F^{\pi_k}(x) > 0] A_F^{\pi_k}(x, u)\right] = 0.$$

By Theorem 2, we have

$$F^{\pi_{k+1}}(x) \le F^{\pi_k}(x) + \frac{1}{1-\gamma}\mathbb{E}_{x' \sim d_0^{\pi_k}(\cdot|x), u' \sim \pi_{k+1}}\left[A_F^{\pi_k}(x', u')\right]$$
$$+ \frac{2\gamma \epsilon_F^{\pi_{k+1}}}{(1-\gamma)^2}\mathbb{E}_{x' \sim d^{\pi_k}(\cdot|x)}\left[D_{TV}(\pi_{k+1} \| \pi_k)[x']\right]. \tag{22}$$

For all $x \in \mathrm{X}^{\pi_k}$ and all $u \in \mathcal{U}$ such that $\pi_{k+1}(u|x) > 0$, we have

$$F^{\pi_k}(x) + A_F^{\pi_k}(x, u)/(1-\gamma) \le 0 \Rightarrow A_F^{\pi_k}(x, u) \le 0.$$

Take expectations inside the feasible region on both sides of (22),

$$\mathbb{E}_{x \sim d_{\text{init}}}[\mathbb{I}[F^{\pi_k}(x) \le 0] F^{\pi_{k+1}}(x)] \le \mathbb{E}_{x \sim d_{\text{init}}}\left[\mathbb{I}[F^{\pi_k}(x) \le 0] F^{\pi_k}(x)\right.$$
$$+ \mathbb{E}_{x \sim d_0^{\pi_k}, u \sim \pi_{k+1}}\left[\mathbb{I}[F^{\pi_k}(x) \le 0] A_F^{\pi_k}(x, u)\right]$$
$$+ \frac{2\gamma \epsilon_F^{\pi_{k+1}}}{(1-\gamma)^2}\mathbb{E}_{x \sim d^{\pi_k}}\left[\mathbb{I}[F^{\pi_k}(x) \le 0] D_{TV}(\pi_{k+1} \| \pi_k)[x]\right]$$
$$\le \frac{2\gamma \epsilon_F^{\pi_{k+1}}}{(1-\gamma)^2}\mathbb{E}_{x \sim d^{\pi_k}}\left[\mathbb{I}[F^{\pi_k}(x) \le 0] D_{TV}(\pi_{k+1} \| \pi_k)[x]\right].$$

Using the relationship $D_{TV}(p\|q) \le \sqrt{D_{KL}(p\|q)/2}$ and Jensen's inequality, we have

$$\mathbb{E}_{x \sim d_{\text{init}}}[\mathbb{I}[F^{\pi_k}(x) \le 0] F^{\pi_{k+1}}(x)] \le \frac{2\gamma \epsilon_F^{\pi_{k+1}}}{(1-\gamma)^2}\mathbb{E}_{x \sim d^{\pi_k}}\left[\mathbb{I}[F^{\pi_k}(x) \le 0]\sqrt{D_{KL}(\pi_{k+1}\|\pi_k)[x]/2}\right]$$
$$\le \frac{2\gamma \epsilon_F^{\pi_{k+1}}}{(1-\gamma)^2}\sqrt{\mathbb{E}_{x \sim d^{\pi_k}}[\mathbb{I}[F^{\pi_k}(x) \le 0] D_{KL}(\pi_{k+1}\|\pi_k)[x]]/2}$$
$$\le \frac{2\gamma \epsilon_F^{\pi_{k+1}}}{(1-\gamma)^2} \cdot \sqrt{\delta/4}$$
$$= \frac{\sqrt{\delta}\gamma \epsilon_F^{\pi_{k+1}}}{(1-\gamma)^2},$$

which proves the first inequality of the feasibility enhancement property.

Rearrange (22) and take expectations outside the feasible region, we have

$$\mathbb{E}_{x \sim d_{\text{init}}}[\mathbb{I}[F^{\pi_k}(x) > 0](F^{\pi_{k+1}}(x) - F^{\pi_k}(x)]$$
$$\leq \mathbb{E}_{x \sim d_0^{\pi_k}, u \sim \pi_{k+1}}[\mathbb{I}[F^{\pi_k}(x) > 0]A_F^{\pi_k}(x, u)]$$
$$+ \frac{2\gamma\epsilon_F^{\pi_{k+1}}}{(1-\gamma)^2}\mathbb{E}_{x \sim d^{\pi_k}}[\mathbb{I}[F^{\pi_k}(x) > 0]D_{TV}(\pi_{k+1}\|\pi_k)[x]]$$
$$\leq \frac{2\gamma\epsilon_F^{\pi_{k+1}}}{(1-\gamma)^2}\mathbb{E}_{x \sim d^{\pi_k}}[\mathbb{I}[F^{\pi_k}(x) > 0]D_{TV}(\pi_{k+1}\|\pi_k)[x]]$$
$$\leq \frac{2\gamma\epsilon_F^{\pi_{k+1}}}{(1-\gamma)^2}\sqrt{\mathbb{E}_{x \sim d^{\pi_k}}[\mathbb{I}[F^{\pi_k}(x) > 0]D_{KL}(\pi_{k+1}\|\pi_k)[x]]/2}$$
$$\leq \frac{\sqrt{\delta}\gamma\epsilon_F^{\pi_{k+1}}}{(1-\gamma)^2}.$$

This proves the second inequality of the feasibility enhancement property.

By Corollary 1 in the CPO paper (Achiam et al., 2017), we have

$$V^{\pi_{k+1}}(x) - V^{\pi_k}(x) \geq \frac{1}{1-\gamma}\mathbb{E}_{x' \sim d^{\pi_k}(\cdot|x), u' \sim \pi_{k+1}}[A^{\pi_k}(x', u')]$$
$$- \frac{2\gamma\epsilon^{\pi_{k+1}}}{(1-\gamma)^2}\mathbb{E}_{x' \sim d^{\pi_k}(\cdot|x)}[D_{TV}(\pi_{k+1}\|\pi_k)[x']]. \tag{23}$$

Take expectations inside the feasible region,

$$\mathbb{E}_{x \sim d_{\text{init}}}[\mathbb{I}[F^{\pi_k}(x) \leq 0](V^{\pi_{k+1}}(x) - V^{\pi_k}(x))]$$
$$\geq \frac{1}{1-\gamma}\mathbb{E}_{x \sim d^{\pi_k}}[\mathbb{I}[F^{\pi_k}(x) \leq 0]A^{\pi_k}(x, u)]$$
$$- \frac{2\gamma\epsilon^{\pi_{k+1}}}{(1-\gamma)^2}\mathbb{E}_{x \sim d^{\pi_k}}[\mathbb{I}[F^{\pi_k}(x) \leq 0]D_{TV}(\pi_{k+1}\|\pi_k)[x]]$$
$$\geq - \frac{2\gamma\epsilon^{\pi_{k+1}}}{(1-\gamma)^2}\mathbb{E}_{x \sim d^{\pi_k}}[\mathbb{I}[F^{\pi_k}(x) \leq 0]D_{TV}(\pi_{k+1}\|\pi_k)[x]]$$
$$\geq - \frac{\sqrt{\delta}\gamma\epsilon^{\pi_{k+1}}}{(1-\gamma)^2}.$$

This proves the value improvement property and thus finishes the proof. □

## B  PRACTICAL IMPLEMENTATION

### B.1  DERIVATION OF GAE OF CDF

For a given trajectory $x_1, x_2, x_3, \ldots$, define

$$\delta_{F,t} = c_t + (1 - c_t)\gamma F(x_{t+1}) - F(x_t).$$

Consider the multi-step TD errors of the CDF up to $k$ steps:

$$A_F^{(1)} = \delta_{F,t}$$
$$= -F(x_t) + c_t + (1 - c_t)\gamma F(x_{t+1}),$$
$$A_F^{(2)} = \delta_{F,t} + (1 - c_t)\gamma\delta_{F,t+1}$$
$$= c_t + (1 - c_t)\gamma(c_{t+1} + (1 - c_{t+1})\gamma F(x_{t+2})) - F(x_t)$$
$$= -F(x_t) + c_t + (1 - c_t)c_{t+1}\gamma + (1 - c_t)(1 - c_{t+1})\gamma^2 F(x_{t+2}),$$

$$\vdots$$

$$A_F^{(k)} = \sum_{l=0}^{k-1} \gamma^l \prod_{s=0}^{l-1} (1 - c_{t+s})\, \delta_{F,t+l}$$
$$= -F(x_t) + c_t + (1 - c_t)c_{t+1}\gamma + (1 - c_t)(1 - c_{t+1})c_{t+2}\gamma^2 + \dots$$
$$+ \prod_{s=0}^{k-2} (1 - c_{t+s})\, c_{t+k-1}\gamma^{k-1} + \prod_{s=0}^{k-1} (1 - c_{t+s})\, \gamma^k F(s_{t+k})$$

The GAE of the CDF is the exponentially-weighted average of these $k$-step TD errors:

$$A_F = (1 - \lambda)\left(A_F^{(1)} + \lambda A_F^{(2)} + \lambda^2 A_F^{(3)} + \dots\right)$$

$$= (1 - \lambda)\Bigg(\delta_{F,t} + \lambda(\delta_{F,t} + (1 - c_t)\gamma\delta_{t+1}^F) + \lambda^2(\delta_{F,t} + (1 - c_t)\gamma\delta_{F,t+1}$$

$$+ (1 - c_t)(1 - c_{t+1})\gamma^2\delta_{F,t+2} + \dots)\Bigg)$$

$$= (1 - \lambda)\Bigg(\left(1 + \lambda + \lambda^2 + \dots\right)\delta_{F,t} + \lambda\gamma(1 - c_t)\left(1 + \lambda + \lambda^2 + \dots\right)\delta_{F,t+1}$$

$$+ (\lambda\gamma)^2(1 - c_t)(1 - c_{t+1})\left(1 + \lambda + \lambda^2 + \dots\right)\delta_{F,t+2} + \dots\Bigg)$$

$$= (1 - \lambda)\left(\frac{1}{1 - \lambda}\delta_{F,t} + (1 - c_t)\frac{\lambda\gamma}{1 - \lambda}\delta_{F,t+1} + (1 - c_t)(1 - c_{t+1})\frac{(\lambda\gamma)^2}{1 - \lambda}\delta_{F,t+2} + \dots\right)$$

$$= \sum_{l=0}^{\infty} (\lambda\gamma)^l \prod_{s=0}^{l-1} (1 - c_{t+s})\delta_{F,t+l}.$$

## B.2 PSEUDOCODE

---
**Algorithm 1:** Feasible policy optimization (FPO)

---
**Initialize:** Network parameters $\phi, \omega, \theta$.

**1 for** *each epoch* **do**

    // Sample data

**2**    **for** *each sample step* **do**

**3**      Sample action $u \sim \pi_\theta(\cdot|x)$;

**4**      Get next state $x'$, reward $r$, and indicator for constraint violation $c$ from environment;

**5**    **end**

**6**    Compute GAEs of return and cost along sampled trajectories;

    // Update networks

**7**    **for** *each update step* **do**

**8**      Update feasibility network $\phi \leftarrow \phi - \eta \nabla_\phi L_F(\phi)$;        // Equation (16)

**9**      Update value network $\omega \leftarrow \omega - \eta \nabla_\omega L_V(\omega)$;        // Equation (17)

**10**      Update policy network $\theta \leftarrow \theta - \eta \nabla_\theta L_\pi(\theta)$;        // Equation (18)

**11**      Update weight exponent by Equation (14);

**12**    **end**

**13 end**

---

## C  EXPERIMENTS

The Safety-Gymnasium benchmark (Ji et al., 2023a) and the Omnisafe toolbox (Ji et al., 2023b) are both released under the Apache License 2.0.

All experiments are conducted on a workstation equipped with Intel(R) Xeon(R) Gold 6246R CPUs (32 cores, 64 threads), an NVIDIA GeForce RTX 3090 GPU, and 256GB of RAM. A single experimental trial—comprising one environment, one algorithm, and one random seed—takes about 2 hours to execute. Executing all experiments with a properly configured concurrent running scheme requires approximately 400 hours.

### C.1  HYPERPARAMETERS

Table 1: Hyperparameters

| Category | Hyperparameter | Value |
|---|---|---|
| Shared | Number of vector environments | 20 |
| | Steps per epoch | 20000 |
| | Batch size | 20000 for navigation tasks |
| | | 4000 for velocity tasks |
| | Reward discount factor | 0.99 |
| | Cost discount factor | 0.95 |
| | Cost limit | 0 |
| | GAE $\lambda$ | 0.95 |
| | Actor learning rate | 3e-5 for PointCircle |
| | | 3e-4 for CarCircle, Ant, HalfCheetah, |
| | | Hopper, and Walker2d |
| | | 1e-4 for others |
| | Actor learning rate schedule | linear decay to 0 |
| | Actor network hidden sizes | (64, 64) |
| | Actor activation function | Tanh |
| | Critic learning rate | 3e-4 |
| | Critic network hidden sizes | (64, 64) |
| | Critic activation function | Tanh |
| | Network weight initialization method | Kaiming uniform |
| | Optimizer | Adam |
| | Entropy coefficient | 0.01 for Hopper and Walker2d |
| | | 0 for others |
| | Critic norm coefficient | 0.001 |
| | Target KL divergence | 0.02 |
| | Maximum gradient norm | 40 |
| PPO | IS ratio clip | 0.2 |
| Lagrangian | Initial multiplier | 0.001 |
| | Multiplier learning rate | 0.035 |
| FPO | Feasibility threshold $\epsilon$ | 0.1 |
| | Initial weight exponent $\beta$ | 0.001 |
| | Weight exponent learning rate | 0.035 |

### C.2  ADDITIONAL RESULTS

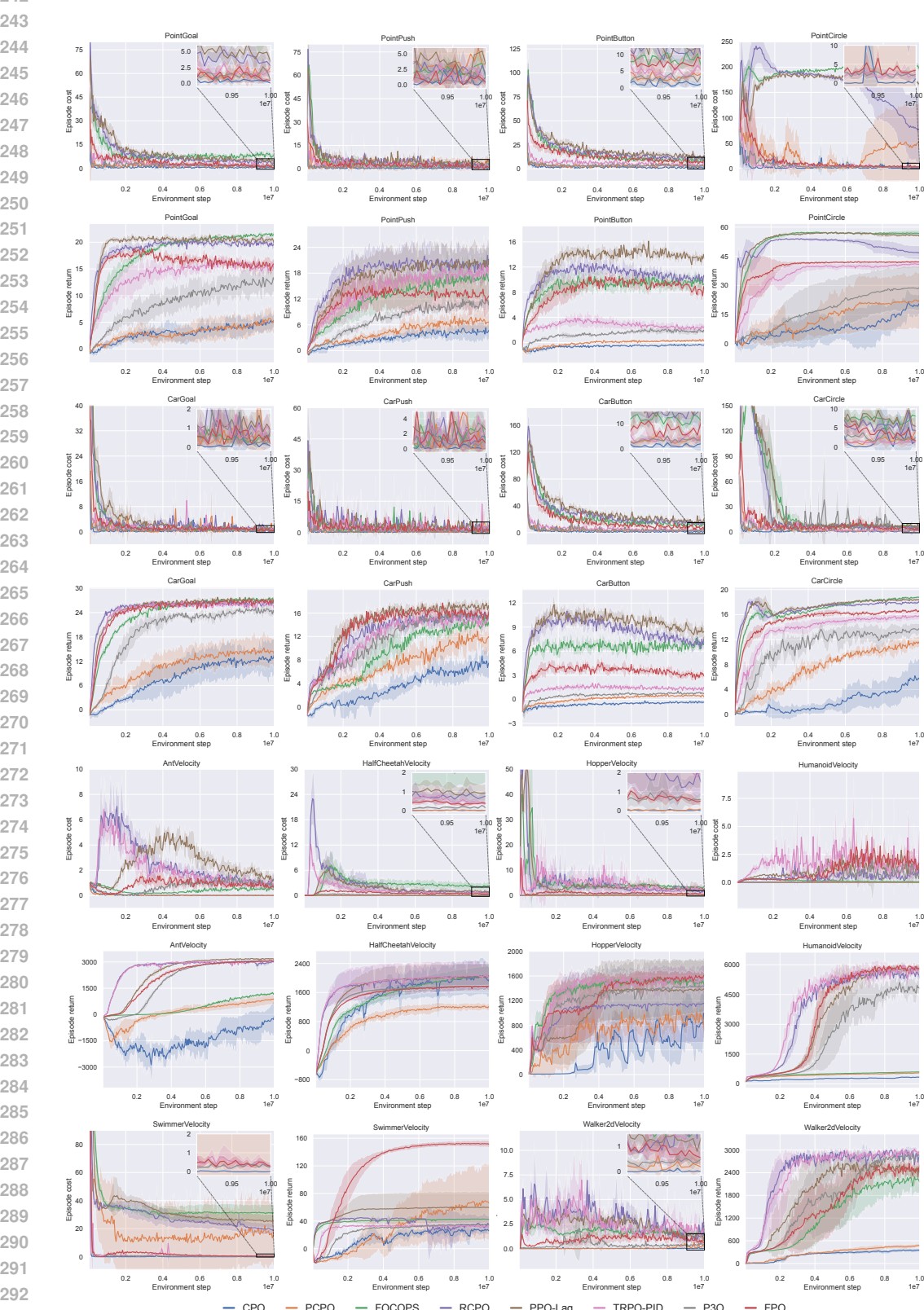

Figure 6: Training curves on all 14 environments in Safety-Gymnasium benchmark. The shaded areas represent 95% confidence intervals over 5 seeds.

Table 2: Average cost and return in the last 10% iterations

| | AntVelocity | | CarButton | | CarCircle | |
|---|---|---|---|---|---|---|
| Algorithm | Cost | Return | Cost | Return | Cost | Return |
| CPO | $0.01 \pm 0.01$ | $-460.19 \pm 740.62$ | $1.22 \pm 0.38$ | $-0.38 \pm 0.18$ | $1.80 \pm 2.51$ | $5.30 \pm 2.65$ |
| PCPO | $0.01 \pm 0.00$ | $823.17 \pm 292.59$ | $3.52 \pm 0.42$ | $0.42 \pm 0.09$ | $1.78 \pm 0.88$ | $11.14 \pm 1.01$ |
| FOCOPS | $0.53 \pm 0.07$ | $1147.42 \pm 69.22$ | $11.83 \pm 0.67$ | $6.78 \pm 0.22$ | $6.98 \pm 1.12$ | $18.68 \pm 0.24$ |
| RCPO | $1.00 \pm 0.46$ | $2942.00 \pm 145.65$ | $15.00 \pm 1.72$ | $7.13 \pm 0.41$ | $5.48 \pm 0.80$ | $17.89 \pm 0.20$ |
| PPO-Lag | $1.43 \pm 0.36$ | $3172.45 \pm 61.49$ | $16.49 \pm 1.99$ | $8.56 \pm 0.58$ | $6.86 \pm 1.14$ | $18.34 \pm 0.25$ |
| TRPO-PID | $0.97 \pm 0.20$ | $3050.21 \pm 37.43$ | $3.96 \pm 0.45$ | $1.35 \pm 0.09$ | $1.26 \pm 0.43$ | $15.66 \pm 0.60$ |
| P3O | $0.98 \pm 0.17$ | $3065.16 \pm 18.30$ | $3.48 \pm 1.14$ | $0.76 \pm 0.09$ | $2.55 \pm 0.38$ | $13.46 \pm 1.05$ |
| ASCPO | $0.43 \pm 0.05$ | $216.45 \pm 12.61$ | $174.63 \pm 22.76$ | $2.03 \pm 0.13$ | $132.78 \pm 8.28$ | $10.37 \pm 0.06$ |
| FPO | $0.76 \pm 0.18$ | $3039.96 \pm 32.59$ | $7.54 \pm 0.69$ | $3.06 \pm 0.19$ | $3.73 \pm 0.99$ | $16.55 \pm 0.24$ |

| | CarGoal | | CarPush | | HalfCheetahVelocity | |
|---|---|---|---|---|---|---|
| Algorithm | Cost | Return | Cost | Return | Cost | Return |
| CPO | $0.15 \pm 0.13$ | $12.52 \pm 4.26$ | $0.31 \pm 0.36$ | $7.31 \pm 1.96$ | $0.01 \pm 0.01$ | $2008.63 \pm 478.16$ |
| PCPO | $0.55 \pm 0.29$ | $14.43 \pm 3.71$ | $1.10 \pm 0.79$ | $11.56 \pm 2.02$ | $0.01 \pm 0.01$ | $1201.48 \pm 97.06$ |
| FOCOPS | $0.48 \pm 0.04$ | $27.18 \pm 0.16$ | $1.54 \pm 1.03$ | $14.21 \pm 0.55$ | $2.18 \pm 0.88$ | $2047.42 \pm 257.71$ |
| RCPO | $1.11 \pm 0.18$ | $26.27 \pm 0.35$ | $1.98 \pm 0.75$ | $15.52 \pm 1.35$ | $0.75 \pm 0.31$ | $2059.44 \pm 394.40$ |
| PPO-Lag | $1.03 \pm 0.39$ | $27.02 \pm 0.12$ | $2.30 \pm 0.25$ | $17.08 \pm 0.64$ | $1.01 \pm 0.48$ | $2020.78 \pm 373.69$ |
| TRPO-PID | $0.81 \pm 0.22$ | $26.07 \pm 0.26$ | $2.27 \pm 1.48$ | $14.78 \pm 1.01$ | $0.56 \pm 0.21$ | $2057.14 \pm 390.16$ |
| P3O | $0.74 \pm 0.41$ | $24.41 \pm 0.82$ | $1.01 \pm 0.91$ | $15.50 \pm 0.71$ | $0.16 \pm 0.03$ | $1770.94 \pm 25.48$ |
| ASCPO | $52.15 \pm 0.93$ | $13.70 \pm 0.38$ | $31.90 \pm 1.95$ | $4.13 \pm 0.25$ | $8.48 \pm 6.11$ | $711.80 \pm 98.35$ |
| FPO | $0.68 \pm 0.27$ | $26.81 \pm 0.17$ | $1.36 \pm 0.32$ | $15.90 \pm 0.61$ | $0.41 \pm 0.07$ | $1764.49 \pm 26.14$ |

| | HopperVelocity | | HumanoidVelocity | | PointButton | |
|---|---|---|---|---|---|---|
| Algorithm | Cost | Return | Cost | Return | Cost | Return |
| CPO | $0.01 \pm 0.01$ | $808.94 \pm 258.62$ | $0.01 \pm 0.00$ | $334.46 \pm 39.26$ | $1.29 \pm 0.25$ | $-0.37 \pm 0.05$ |
| PCPO | $0.03 \pm 0.05$ | $923.05 \pm 77.47$ | $0.00 \pm 0.00$ | $539.65 \pm 25.04$ | $3.22 \pm 0.74$ | $0.32 \pm 0.20$ |
| FOCOPS | $3.31 \pm 0.62$ | $1502.24 \pm 63.07$ | $0.08 \pm 0.01$ | $594.57 \pm 35.36$ | $8.69 \pm 1.31$ | $9.63 \pm 1.18$ |
| RCPO | $1.97 \pm 1.07$ | $1139.05 \pm 618.61$ | $0.63 \pm 0.19$ | $5555.85 \pm 218.92$ | $11.52 \pm 0.78$ | $10.24 \pm 0.34$ |
| PPO-Lag | $2.89 \pm 1.70$ | $1376.34 \pm 501.10$ | $1.43 \pm 0.29$ | $5742.21 \pm 250.88$ | $12.38 \pm 0.61$ | $13.43 \pm 0.37$ |
| TRPO-PID | $2.57 \pm 1.61$ | $1531.96 \pm 183.29$ | $1.72 \pm 0.86$ | $5706.46 \pm 197.10$ | $4.76 \pm 0.58$ | $2.34 \pm 0.62$ |
| P3O | $0.66 \pm 0.54$ | $1429.71 \pm 430.99$ | $1.05 \pm 0.75$ | $4792.15 \pm 400.03$ | $3.02 \pm 0.51$ | $1.79 \pm 0.42$ |
| ASCPO | $6.40 \pm 1.67$ | $27.65 \pm 6.25$ | $0.00 \pm 0.00$ | $85.42 \pm 12.98$ | $87.63 \pm 3.64$ | $2.60 \pm 0.41$ |
| FPO | $0.67 \pm 0.18$ | $1572.20 \pm 92.16$ | $1.83 \pm 0.47$ | $5842.85 \pm 75.03$ | $7.36 \pm 1.12$ | $8.48 \pm 0.74$ |

| | PointCircle | | PointGoal | | PointPush | |
|---|---|---|---|---|---|---|
| Algorithm | Cost | Return | Cost | Return | Cost | Return |
| CPO | $1.26 \pm 2.18$ | $19.09 \pm 6.21$ | $0.40 \pm 0.22$ | $4.96 \pm 1.61$ | $0.72 \pm 0.54$ | $4.41 \pm 2.12$ |
| PCPO | $46.57 \pm 78.77$ | $21.76 \pm 14.25$ | $1.63 \pm 0.66$ | $5.05 \pm 2.11$ | $3.08 \pm 3.15$ | $6.61 \pm 2.72$ |
| FOCOPS | $200.57 \pm 7.47$ | $56.83 \pm 1.50$ | $8.29 \pm 1.08$ | $21.38 \pm 0.15$ | $2.14 \pm 0.58$ | $16.80 \pm 4.79$ |
| RCPO | $85.71 \pm 60.89$ | $47.23 \pm 3.35$ | $4.12 \pm 0.62$ | $19.77 \pm 0.31$ | $2.78 \pm 1.41$ | $19.84 \pm 4.02$ |
| PPO-Lag | $170.81 \pm 6.63$ | $55.85 \pm 0.85$ | $5.05 \pm 0.98$ | $20.44 \pm 0.54$ | $3.87 \pm 1.05$ | $20.01 \pm 4.59$ |
| TRPO-PID | $2.43 \pm 0.96$ | $40.84 \pm 2.10$ | $1.82 \pm 0.39$ | $15.24 \pm 1.08$ | $1.53 \pm 0.27$ | $18.02 \pm 0.98$ |
| P3O | $2.42 \pm 0.80$ | $28.54 \pm 11.22$ | $1.39 \pm 0.38$ | $12.44 \pm 2.46$ | $0.97 \pm 0.33$ | $10.90 \pm 1.16$ |
| ASCPO | $139.85 \pm 4.59$ | $23.30 \pm 1.41$ | $57.72 \pm 0.68$ | $10.12 \pm 0.96$ | $37.19 \pm 3.30$ | $8.33 \pm 1.07$ |
| FPO | $3.67 \pm 0.68$ | $42.03 \pm 0.62$ | $1.26 \pm 0.07$ | $15.53 \pm 0.96$ | $1.10 \pm 0.40$ | $12.86 \pm 3.78$ |

| | SwimmerVelocity | | Walker2dVelocity | |
|---|---|---|---|---|
| Algorithm | Cost | Return | Cost | Return |
| CPO | $0.02 \pm 0.01$ | $26.99 \pm 6.05$ | $0.02 \pm 0.02$ | $348.08 \pm 48.28$ |
| PCPO | $17.71 \pm 24.39$ | $66.14 \pm 53.32$ | $0.22 \pm 0.21$ | $454.43 \pm 54.78$ |
| FOCOPS | $31.33 \pm 6.06$ | $42.82 \pm 1.45$ | $1.56 \pm 0.48$ | $2200.94 \pm 324.12$ |
| RCPO | $19.86 \pm 4.13$ | $34.31 \pm 14.90$ | $1.26 \pm 0.25$ | $2849.69 \pm 109.22$ |
| PPO-Lag | $25.53 \pm 4.87$ | $59.85 \pm 22.59$ | $1.28 \pm 0.53$ | $2489.10 \pm 196.43$ |
| TRPO-PID | $0.54 \pm 0.23$ | $32.46 \pm 2.75$ | $1.84 \pm 1.11$ | $2927.93 \pm 55.02$ |
| P3O | $0.29 \pm 0.01$ | $34.92 \pm 1.56$ | $0.39 \pm 0.06$ | $2823.77 \pm 145.66$ |
| ASCPO | $85.12 \pm 44.63$ | $-3.90 \pm 7.65$ | $0.18 \pm 0.20$ | $1.01 \pm 4.62$ |
| FPO | $0.44 \pm 0.17$ | $152.10 \pm 4.67$ | $0.84 \pm 0.29$ | $2481.97 \pm 217.58$ |

## D   LARGE LANGUAGE MODEL USAGE DISCLOSURE

We used Large Language Model (LLM) solely for the purpose of improving grammar and polishing writing. The LLM was not used for any core research tasks such as retrieval, discovery, ideation, or analysis.

