# OpenReview forum: "Feasible Policy Optimization for Safe Reinforcement Learning"
_ICLR.cc/2026/Conference — Submitted to ICLR 2026_

### Official Review · Reviewer_6eAt · 2025-10-26

**Soundness:** 3
**Presentation:** 3
**Contribution:** 3
**Rating:** 6
**Confidence:** 4

**Summary:**

The paper talks about an efficient approach where, instead of enforcing the constraint satisfaction explicitly after every policy update, it talks about maximizing the Value function in the present feasible domain and then progressively expanding the feasible domain.
In particular, the paper solves the problem in two ways. At first, they solve
$\max_{\pi} \mathbb{E}_{x \sim d_{init}}\left[ \mathbb{I}\left[{F^{\pi_{k}} \leq 0}\right]V^{\pi}(x)\right]$
s.t. $ \max_{\pi} \mathbb{E}_{x \sim d_{init}}\left[ \mathbb{I}\left[{F^{\pi_{k}} \leq 0}\right]F^{\pi}(x)\right] \leq 0$

Simultaneously, they aim to expand the feasible range by solving the following optimization problem.
$\max_{\pi} \mathbb{E}_{x \sim d_{init}}\left[ \mathbb{I}\left[{F^{\pi_{k}} > 0}\right]F^{\pi}(x)\right]$
s.t. $ \max_{\pi} \mathbb{E}_{x \sim d_{init}}\left[ \mathbb{I}\left[{F^{\pi_{k}} \leq 0}\right]F^{\pi}(x)\right] \leq 0$

They show that the two problems have a common optimization solution and further finds the feasibility bounds of their solution.

**Strengths:**

1. The Constraint enforcement at every step is shown to be overly conservative and not necessary
2. Talks about a Feasible Policy Optimization algorithm which aims to get the safe policy at every instant
3. At each step it can guarantee policy improvement almost surely.
4. Evaluated on some standard Safe RL benchmarks.

**Weaknesses:**

1. The writing was little hard to follow with a lot of minor gaps in equations for example, In equation 11 $k$,  should come in the subscript for the equations. $d^{\pi_{k}}$ and not $d^\pi k$ and  if $A^{\pi_{k}}(x,u)$ was introduced

2. The reachability set is little too stringent. I get that it is required for the proofs but it created confusions too. For example. In Theorem 1 the paper claims that if $\pi_{k+1}$ gets a value in $$ \mathcal{R}^{\pi_{in}(\mathcal{X}_{init} \cap X^{\pi_{k}}) $$, $\pi_{k+1}$ would be replaced by $\pi_{in}$. This need not be true always. This is because $\mathcal{R}$ is a reability state. And according to definition of $\mathcal{R}$ introduced in the paper, it might be possible that $\pi_{k+1}$ and $\pi_{k}$ can have some overlapping reachability sets but it does not necessarily means that it will switch to some other policy.

3. Over usage of $c$, used as cost as well as $\mathbb{I}\left[ h(x) > 0\right]$ for indicator constraint violation. Is the constraint violation same as cost or not. These small details made it little hard to follow,

**Questions:**

1. The reachability set is little too stringent. I get that it is required for the proofs but it created confusions too. For example. In Theorem 1 the paper claims that if $\pi_{k+1}$ gets a value in $$ \mathcal{R}^{\pi_{in}(\mathcal{X}_{init} \cap X^{\pi_{k}}) $$, $\pi_{k+1}$ would be replaced by $\pi_{in}$. This need not be true always. This is because $\mathcal{R}$ is a reability state. And according to definition of $\mathcal{R}$ introduced in the paper, it might be possible that $\pi_{k+1}$ and $\pi_{k}$ can have some overlapping reachability sets but it does not necessarily means that it will switch to some other policy. It would be helpful if this is explicitly clarified.

2. In Figure 1, we observe that the return and cost of FPO are the best, which is great. I am just curious to know that only FPO has the lowest uncertainty range in terms of cost. But its uncertainty range in terms of return is a little higher. So, if we run for long enough as you have for your experiments, why is the uncertainty range for other algorithms, such as PPO-Lag and FOCOPS, so vast? Because for PPO-Lag, there is a possibility that it can go less than 0.08 in cost while giving better value return. For example, if you check the Point Button plot, PPO-lag has supremely outperformed FPO.

3. Also curious how the given FPO algorithm would perform compared to CRPO

---

> ### Author Response · Authors · 2025-11-21
> **Authors' Response (Part 1)**
>
> We sincerely appreciate the reviewer's time and effort in reviewing our paper and providing such constructive feedback. Below, we address each of the concerns in detail.
>
> ## Weaknesses
>
> > The writing was a little hard to follow with many minor gaps in equations; for example, in Eq. (11) the $k$ should appear in the subscript. One expects $d^{\pi_{k}}$ (and not $d^{\pi}k$), and if $A^{\pi_{k}}(x,u)$ was introduced, it should be used consistently.
>
> We thank the reviewer for their careful reading. The notation in Eq. (11) is indeed $d^{\pi_{k}}$, not $d^{\pi}k$. The confusion likely arose due to the subscript formatting in the template making the notation appear ambiguous. Regarding $A^{\pi_{k}}(x,u)$, the reviewer is correct that it was not explicitly defined earlier. It denotes the standard advantage function in RL. We have revised the manuscript to include its definition and ensure notational consistency throughout.
>
> > The reachability set is a little too stringent. I get that it is required for the proofs but it created confusions too. For example, in Theorem 1 the paper claims that if $\pi_{k+1}$ gets a value in $\mathcal{R}^{\pi_\text{in}}(X_\text{init} \cap X^{\pi_k})$, then $\pi_{k+1}$ would be replaced by $\pi_{\text{in}}$. This need not be true always. This is because $\mathcal{R}$ is a reachability set. According to the definition of $\mathcal{R}$ introduced in the paper, it might be possible that $\pi_{k+1}$ and $\pi_k$ have overlapping reachability sets, but that does not necessarily mean it will switch to some other policy.
>
> We appreciate the reviewer’s insightful observation. The reviewer is correct that the policy need not switch in the overlapping part of the reachable sets of $\pi_{k+1}$ and $\pi_k$. In fact, Theorem 1 does not claim that a switch must occur, but only requires that a policy exists which is the optimal solution to both problems. The construction in Eq. (9) provides one such valid policy. The existence of other valid policies, like remaining with $\pi_k$ in the overlap, does not invalidate the theorem, as it is an existential ("there exists") rather than a universal ("for all") claim. We have added an explanation after Theorem 1 to clarify this.
>
> > Over-usage of $c$: it is used as the cost as well as the indicator $\mathbb{I}\left[h(x)>0\right]$ for constraint violation. Is the constraint violation the same as cost or not? These small details made it a little hard to follow.
>
> We thank the reviewer for highlighting this ambiguity. The symbol $c$ is intended to solely represents the indicator for constraint violation, and not an additional cost function. We have revised the pseudocode in Appendix B.2 to consistently refer to it as the "indicator for constraint violation" to eliminate confusion.

---

> ### Author Response · Authors · 2025-11-21
> **Authors' Response (Part 2)**
>
> ## Questions
>
> > The reachability set is a little too stringent. I get that it is required for the proofs but it created confusions too. For example, in Theorem 1 the paper claims that if $\pi_{k+1}$ gets a value in $\mathcal{R}^{\pi_{\text{in}}}(X_\text{init} \cap X^{\pi_k})$, then $\pi_{k+1}$ would be replaced by $\pi_{\text{in}}$. This need not be true always. This is because $\mathcal{R}$ is a reachability set. According to the definition of $\mathcal{R}$ introduced in the paper, it might be possible that $\pi_{k+1}$ and $\pi_k$ have overlapping reachability sets, but that does not necessarily mean it will switch to some other policy. It would be helpful if this is explicitly clarified.
>
> See our response to Weakness 2.
>
> > In Figure 1, we observe that the return and cost of FPO are the best, which is great. I am just curious to know that only FPO has the lowest uncertainty range in terms of cost. But its uncertainty range in terms of return is a little higher. So, if we run for long enough as you have for your experiments, why is the uncertainty range for other algorithms, such as PPO-Lag and FOCOPS, so vast? Because for PPO-Lag, there is a possibility that it can go less than 0.08 in cost while giving better value return. For example, if you check the Point Button plot, PPO-lag has supremely outperformed FPO.
>
> We thank the reviewer for this insightful question. The reason that the uncertainty ranges of algorithms like PPO-Lag and FOCOPS are large is that their scores vary substantially across tasks. This variability is intrinsic and cannot be eliminated by longer training. In contrast, FPO achieves more consistent performance across tasks and thus has smaller uncertainty range. It is worth mentioning that FPO uses a single set of hyperparameters for all tasks, demonstrating its strong generalization ability. The slightly higher uncertainty in FPO's return arises because, once cost is very low, further reducing it may require larger trade-offs in return. The extent of this trade-off differs per environment, leading to greater variance in return. A similar pattern is observed in other low-cost algorithms like TRPO-PID and P3O.
>
> Regarding the possibility of PPO‑Lag achieving lower cost with better return, we performed an ablation where we increased its Lagrange multiplier (initial value and learning rate) by 3×:
>
> |Environment|PPO-Lag||PPO-Lag (3x)||FPO||
> |-|-|-|-|-|-|-|
> ||Cost|Return|Cost|Return|Cost|Return|
> |PointGoal|0.096|0.851|0.044|0.789|0.024|0.647|
> |PointPush|0.100|0.658|0.074|0.663|0.028|0.423|
> |PointButton|0.106|0.476|0.059|0.251|0.063|0.300|
> |PointCircle|0.756|1.098|0.017|0.855|0.016|0.827|
> |CarGoal|0.019|0.893|0.017|0.887|0.012|0.886|
> |CarPush|0.126|0.908|0.109|0.884|0.074|0.846|
> |CarButton|0.079|0.306|0.039|0.130|0.036|0.109|
> |CarCircle|0.033|0.772|0.011|0.741|0.018|0.697|
> |AntVelocity|0.002|0.538|0.001|0.540|0.001|0.515|
> |HumanoidVelocity|0.002|0.882|0.000|0.853|0.002|0.898|
> |HalfCheetahVelocity|0.003|0.988|0.001|0.835|0.001|0.863|
> |HopperVelocity|0.008|0.858|0.001|0.707|0.002|0.980|
> |Walker2dVelocity|0.001|0.550|0.000|0.173|0.001|0.549|
> |SwimmerVelocity|0.160|0.498|0.014|0.331|0.003|1.265|
> |**Average**|0.106|0.734|0.028|0.617|0.020|0.700|
>
> The results show that although cost decreases in some tasks (e.g., PointButton), the return is significantly compromised, and the overall performance (average normalized score) becomes inferior to FPO. This indicates that simple hyperparameter tuning is insufficient for PPO-Lag to surpass FPO.
>
> > Also curious how the given FPO algorithm would perform compared to CRPO.
>
> We thank the reviewer for this question. We have conducted experiments comparing FPO with CRPO, and results show that FPO outperforms CRPO in terms of both cost and return.
>
> |Environment|CRPO||FPO||
> |-|-|-|-|-|
> ||Cost|Return|Cost|Return|
> |PointGoal|0.060|0.710|0.024|0.647|
> |PointPush|0.086|0.732|0.028|0.423|
> |PointButton|0.038|0.107|0.063|0.300|
> |PointCircle|0.015|0.884|0.016|0.827|
> |CarGoal|0.022|0.872|0.012|0.886|
> |CarPush|0.211|0.860|0.074|0.846|
> |CarButton|0.028|0.072|0.036|0.109|
> |CarCircle|0.013|0.718|0.018|0.697|
> |AntVelocity|0.001|0.504|0.001|0.515|
> |HumanoidVelocity|0.003|0.803|0.002|0.898|
> |HalfCheetahVelocity|0.005|0.831|0.001|0.863|
> |HopperVelocity|0.006|0.357|0.002|0.980|
> |Walker2dVelocity|0.002|0.552|0.001|0.549|
> |SwimmerVelocity|0.011|0.322|0.003|1.265|
> |**Average**|0.036|0.595|0.020|0.700|

---

### Official Review · Reviewer_KU9E · 2025-10-31

**Soundness:** 3
**Presentation:** 4
**Contribution:** 4
**Rating:** 8
**Confidence:** 3

**Summary:**

The paper proposes a new policy optimization scheme for safe reinforcement learning based on limiting trajectory-wise constraint violations rather than expected constraint violations. The authors develop a dynamic programming method to estimate policy feasibility and combine it with PPO, achieving a scalable RL algorithm that maintains high rewards while avoiding unsafe states.

**Strengths:**

- Novel safety framework introducing the constraint decay function $ F $
- Focus on trajectory-wise feasibility rather than expected feasibility (even though $ F $ is estimated in expectation, it measures violations more directly)
- Clear writing and presentation
- Excellent Figure 1 (should be standard for safe RL)

**Weaknesses:**

- More recent baselines would strengthen the empirical comparison (beyond Ji et al., 2023)
- Curiosity: isn’t $ F $ typically underestimated due to truncated trajectories? Could this affect feasibility estimation?

**Questions:**

- See weak points.

---

> ### Author Response · Authors · 2025-11-21
> **Authors' Response**
>
> We sincerely appreciate the reviewer's time and effort in reviewing our paper and providing such constructive feedback. Below, we address each of the concerns in detail.
>
> ## Weaknesses
>
> > More recent baselines would strengthen the empirical comparison (beyond Ji et al., 2023).
>
> We thank the reviewer for this suggestion. We have added a recently proposed safe RL algorithm, ASCPO (2024) [1], which also addresses the state-wise constraint setting. Initial results (based on one seed) show that FPO outperforms ASCPO in terms of both cost and return. We are currently gathering results over multiple seeds and will include them in the paper once they are completed.
>
> |Environment|ASCPO||FPO||
> |-|-|-|-|-|
> ||Cost|Return|Cost|Return|
> |PointGoal|0.06|0.07|0.02|0.65|
> |PointPush|0.15|-0.03|0.03|0.42|
> |PointButton|0.08|-0.04|0.06|0.30|
> |PointCircle|0.12|0.03|0.02|0.83|
> |CarGoal|0.06|-0.00|0.01|0.89|
> |CarPush|0.34|-0.00|0.07|0.85|
> |CarButton|0.09|-0.05|0.04|0.11|
> |CarCircle|0.04|0.02|0.02|0.70|
> |AntVelocity|0.00|-0.04|0.00|0.52|
> |HalfCheetahVelocity|0.00|0.09|0.00|0.86|
> |HopperVelocity|0.00|0.18|0.00|0.98|
> |HumanoidVelocity|0.00|0.03|0.00|0.90|
> |SwimmerVelocity|0.02|-0.15|0.00|1.27|
> |Walker2dVelocity|0.00|0.01|0.00|0.55|
> |**Average**|0.07|0.01|0.02|0.70|
>
> [1] Zhao, W., et al. (2024). Absolute State-wise Constrained Policy Optimization: High-Probability State-wise Constraints Satisfaction. arXiv preprint arXiv:2410.01212.
>
> > Curiosity: isn’t $F$ typically underestimated due to truncated trajectories? Could this affect feasibility estimation?
>
> We thank the reviewer for this question. We would like to clarify that truncated trajectories do not lead to underestimation of $F$. When computing the GAE for the CDF, we do not set the terminal value of a truncated trajectory to zero, nor do we disregard it. Instead, we use the feasibility value $F(x)$ of the final state reached in the truncated trajectory as the bootstrap value. This ensures that our estimation remains unbiased with respect to trajectory truncation.

---

> ### Comment · Reviewer_KU9E · 2025-11-27
>
> I thank the authors for responding to our review. I feel that the authors have addressed the questions in their rebuttal. I will stick to my positive rating. Thanks a lot.

---

> ### Author Response · Authors · 2025-12-01
>
> Thank you for your positive feedback. The results for ASCPO are included in Table 2 in Appendix C.2.

---

### Official Review · Reviewer_92qC · 2025-11-01

**Soundness:** 2
**Presentation:** 2
**Contribution:** 2
**Rating:** 4
**Confidence:** 4

**Summary:**

This paper propose feasible policy optimization to avoid over-conservatism in safe RL problem. Specifically, it progressively increases feasible region when updating the value function by maximizing the value function in feasible region and minimizing the feasibility function outside. The experiments on safety gymnasium benchmark show that it achieves better balance between reward and cost constraint.

**Strengths:**

- The idea of this paper, i.e., update feasibility function and value function, is clearly presented.
- The topic of this paper is to the interest of safe RL community.

**Weaknesses:**

From the experiment (e.g, in fig.2), it is hard to say FPO is superior to baselines.
- First, the baselines in some tasks are not well set up. For example, PointCircle task is obviously much easier than PointButton and CarGoal. However, the experiments show that PPO-Lag, RCPO, FOCOPS cannot learn a relatively safe policy but they can perform relatively well on PointButton or CarGoal. Actually, I believe the authors did not even try to set a reasonable lagrangian coefficient for them on PointCircle because they perform just like unconstrained RL. As a reference, PPO-Lag can learn well on PointCircle in other library [1].
- Second, the authors claim they follow the original hyperparameter in Omnisafe. However, Omnisafe tuned the hyperparameter for safety constraint $=25$ (and the results show that the baselines can indeed make cost smaller than 25) while this paper uses cost limit $=0$.
- Consider the performances of three hardest tasks (PointButton, CarGoal, CarPush), FPO performs similarly to PPO-Lag and RCPO: FPO has lower reward and a little lower cost.
- The velocity tasks are MUCH easier than navigation tasks in terms of reward-cost trade-off [2]. Meanwhile, the baselines on swimmervelocity is absolutely not well tuned. The ppo-lag and FOCOPS perform well in [1].

**Questions:**

- The authors use $F^\pi(x) \leq 0$ to denote the feasibility region but also use $F=E_\tau[\gamma^{N(\tau)}]$ which is $\geq 0$. So what's the physical meaning of $F^\pi(x) < 0$?
- The compared baselines (e.g., Lagrangian-based) can learn different policies w.r.t different constraint limits. How does FPO adjust to different safety preference?

[1] https://fsrl.readthedocs.io/en/latest/tutorials/benchmark.html

[2] https://safety-gymnasium.readthedocs.io/en/latest/environments/safe_velocity.html#costs

---

> ### Author Response · Authors · 2025-11-21
> **Authors' Response (Part 1)**
>
> We sincerely appreciate the reviewer's time and effort in reviewing our paper and providing such constructive feedback. Below, we address each of the concerns in detail.
>
> ## Weaknesses
>
> > First, the baselines in some tasks are not well set up. For example, PointCircle task is obviously much easier than PointButton and CarGoal. However, the experiments show that PPO-Lag, RCPO, FOCOPS cannot learn a relatively safe policy but they can perform relatively well on PointButton or CarGoal. Actually, I believe the authors did not even try to set a reasonable lagrangian coefficient for them on PointCircle because they perform just like unconstrained RL. As a reference, PPO-Lag can learn well on PointCircle in other library [1].
> >
> > [1] https://fsrl.readthedocs.io/en/latest/tutorials/benchmark.html
>
> We thank the reviewer for this insightful observation. In our experiments, we intentionally used a single, consistent set of hyperparameters for each baseline algorithm across all tasks to test their robustness and ease of use. While it is indeed possible to reduce the cost in PointCircle by aggressively increasing the Lagrangian multiplier, our additional experiments, where we increase both the initial Lagrange multiplier and its learning rate by 3x, show that doing so leads to overly conservative behavior in other environments and inferior overall performance compared to FPO:
>
> |Environment|PPO-Lag||PPO-Lag (3x)||FPO||
> |-|-|-|-|-|-|-|
> ||Cost|Return|Cost|Return|Cost|Return|
> |PointGoal|0.096|0.851|0.044|0.789|0.024|0.647|
> |PointPush|0.100|0.658|0.074|0.663|0.028|0.423|
> |PointButton|0.106|0.476|0.059|0.251|0.063|0.300|
> |PointCircle|0.756|1.098|0.017|0.855|0.016|0.827|
> |CarGoal|0.019|0.893|0.017|0.887|0.012|0.886|
> |CarPush|0.126|0.908|0.109|0.884|0.074|0.846|
> |CarButton|0.079|0.306|0.039|0.130|0.036|0.109|
> |CarCircle|0.033|0.772|0.011|0.741|0.018|0.697|
> |AntVelocity|0.002|0.538|0.001|0.540|0.001|0.515|
> |HumanoidVelocity|0.002|0.882|0.000|0.853|0.002|0.898|
> |HalfCheetahVelocity|0.003|0.988|0.001|0.835|0.001|0.863|
> |HopperVelocity|0.008|0.858|0.001|0.707|0.002|0.980|
> |Walker2dVelocity|0.001|0.550|0.000|0.173|0.001|0.549|
> |SwimmerVelocity|0.160|0.498|0.014|0.331|0.003|1.265|
> |**Average**|0.106|0.734|0.028|0.617|0.020|0.700|
>
> These results indicate that Lagrangian-based methods are highly sensitive to the choice of the multiplier. In contrast, our algorithm achieves robust performance across all environments using the same set of hyperparameters.
>
> > Second, the authors claim they follow the original hyperparameter in Omnisafe. However, Omnisafe tuned the hyperparameter for safety constraint = 25 (and the results show that the baselines can indeed make cost smaller than 25) while this paper uses cost limit = 0.
>
> We thank the reviewer for pointing this out. We agree that stating we followed "the original hyperparameter in Omnisafe" was imprecise. We set the cost limit to zero for all algorithms for a fair comparison, including the Omnisafe baselines, since our objective is to achieve zero constraint violation. We have add an explanation in Section 5.1 to clarify this.
>
> It is important to distinguish the cost limit, which is a problem-level setting that defines the safety objective, from internal algorithmic hyperparameters such as learning rate and batch size. A robust safe RL algorithm should be capable of adapting to different cost limits without requiring an extensive re-tuning of its core hyperparameters.
>
> The most sensitive hyperparameter to changes in the cost limit is typically the Lagrange multiplier. However, as demonstrated in our response to Weakness 1, tuning the multiplier primarily shifts the trade-off between cost and return and is often insufficient to improve the overall performance.

---

> ### Author Response · Authors · 2025-11-21
> **Authors' Response (Part 2)**
>
> > Consider the performances of three hardest tasks (PointButton, CarGoal, CarPush), FPO performs similarly to PPO-Lag and RCPO: FPO has lower reward and a little lower cost.
>
> We appreciate the reviewer's careful observation regarding the three tasks. In our initial assessment, we did not classify these specific tasks as the "hardest" in a standardized sense, as the perceived difficulty can vary depending on the metrics and algorithm. While our evaluation emphasizes overall performance across the suite, we agree that a detailed look at specific tasks is valuable.
>
> For clarity, we have summarized the normalized cost and return for these three tasks below:
>
> |Environment|FPO||PPO-Lag||PPO-Lag (3x)||RCPO||
> |-|-|-|-|-|-|-|-|-|
> ||Cost|Return|Cost|Return|Cost|Return|Cost|Return|
> |PointButton|0.063|0.300|0.106|0.476|0.059|0.251|0.099|0.363|
> |CarGoal|0.012|0.886|0.019|0.893|0.017|0.887|0.020|0.868|
> |CarPush|0.074|0.846|0.126|0.908|0.109|0.884|0.108|0.825|
> |**Average**|0.050|0.677|0.083|0.759|0.062|0.674|0.076|0.685|
>
> From these results, we observe the following:
> - Compared with PPO-Lag, FPO has a lower average return (0.677 vs. 0.759) and also a lower average cost (0.050 vs. 0.083). This is a typical trade-off for safe RL methods.
> - When we increase the PPO-Lag's multiplier by 3x, its cost becomes higher than FPO (0.062 vs. 0.050), and its return becomes slightly lower (0.674 vs. 0.677).
> - RCPO achieves a slightly higher average return than FPO (0.685 vs. 0.677), but at the price of a significantly higher average cost (0.076 vs. 0.050).
>
> Thus, even when focusing on these three tasks, FPO demonstrates a favorable balance between cost and return compared to both PPO-Lag and RCPO. Combined with the aggregate results across all tasks (Fig. 1), we believe this supports the conclusion that FPO achieves excellent safety while maintaining competitive performance.
>
> > The velocity tasks are MUCH easier than navigation tasks in terms of reward-cost trade-off [2]. Meanwhile, the baselines on swimmervelocity is absolutely not well tuned. The ppo-lag and FOCOPS perform well in [1].
> >
> > [2] https://safety-gymnasium.readthedocs.io/en/latest/environments/safe_velocity.html#costs
>
> We thank the reviewer for this comment. We agree that the velocity tasks may present a different, and potentially simpler trade-off than the navigation tasks. However, our experimental goal was to comprehensively evaluate algorithm performance across a diverse suite of environments.
>
> The performance of PPO-Lag and FOCOPS in our experiments on SwimmerVelocity is consistent with their reported results in FSRL [1] and Omnisafe, as shown in the table below. This validates that our baselines are properly configured and the comparison is fair.
>
> |Algorithm|FSRL||Omnisafe||Ours||
> |-|-|-|-|-|-|-|
> ||Cost|Return|Cost|Return|Cost|Return|
> |PPO-Lag|~25|~40|28.02|64.74|25.53|59.85|
> |FOCOPS|~25|~50|29.75|53.87|31.33|42.82|
>
> As analyzed in our response to Weakness 1, while tuning the Lagrange multiplier for PPO-Lag on SwimmerVelocity can further reduce cost, it often sacrifices return. The key advantage of FPO is its ability to achieve robust and consistent performance using a single set of hyperparameters across all tasks.
>
> ## Questions
>
> > The authors use $F^\pi(x)\le0$ to denote the feasibility region but also use $F=E_\tau[\gamma^{N(\tau)}]$ which is $\ge0$. So what's the physical meaning of $F^\pi(x)<0$?
>
> We thank the reviewer for this question. The condition $F^\pi(x)\le0$ is used to define the feasibility region in a general form that accommodates different types of feasibility functions. While the CDF used in our work is non-negative, other feasibility functions, such as the Hamilton-Jacobi reachability value function [3], can take negative values. The notation $F^\pi(x)\le0$ ensures our formulation remains applicable to these broader classes of feasibility functions.
>
> [3] Zheng, Y., et al. (2024). Safe Offline Reinforcement Learning with Feasibility-Guided Diffusion Model. ICLR.
>
> > The compared baselines (e.g., Lagrangian-based) can learn different policies w.r.t different constraint limits. How does FPO adjust to different safety preference?
>
> We thank the reviewer for this important question. Adjusting safety preference is not applicable in our problem setting because we address a fundamentally different problem: state-wise safe RL rather than cumulative constrained RL. Our goal is to achieve state-wise safety, i.e., zero constraint violation at each step, which necessitates a cost limit of zero. The policy should optimize for performance only under the condition of strict safety, rather than trading off safety against performance. This objective aligns with the broad area of state-wise safe RL [4], which differs from the cumulative constraints in CMDP that most baselines address.
>
> [4] Zhao, W., et al. (2023). State-wise safe reinforcement learning: a survey. IJCAI.

---

### Author Response · Authors · 2025-12-03

Dear Area Chair and Reviewers,

Thank you for your engagement in reviewing our paper and for providing the valuable feedback. We are encouraged that two of our three reviewers recommend acceptance, with scores of 8 and 6.

We believe the 4 score from reviewer 92qC is based on a misunderstanding of our baselines configuration. Specifically, reviewer 92qC thinks the hyperparameters of our baselines are not well tuned, and the performance of our baselines is inferior to their reported results. However, we demonstrate in our rebuttal that this is not the case:

- PPO-Lag cannot outperform our algorithm by further tuning the Lagrange multiplier.
- The performance of PPO-Lag and FOCOPS in our paper is consistent with their reported results in other benchmarks.

These results demonstrate that our baselines are already properly configured and the comparison is fair.

In addition, we also improved the paper following other comments from the reviewers:

- KU9E, 6eAt: Add experiments of more recent algorithms.
- 6eAt: Clarify the existential claim of Theorem 1.

We thank you again for your time and consideration.

---

### Meta-Review · Area_Chair_autD · 2026-01-06

**Summary:**

This paper introduces FPO, an algorithm to address the over-conservatism in safe RL with state-wise safety constraints. The key idea is to simultaneously improve the value function and expand the feasible region by solving a region-wise policy optimization problem.

Strengths:
- The idea of introducing the constraint decay function is novel

- The topic of ensuring state-wise safety in RL is important

Weaknesses:
- The empirical performance is not strong enough compared to baselines

**Reviewer Concerns:**

The rebuttal has addressed most concerns raised by the reviewers, including new empirical baselines, comparison with CRPO, and some clarification issues.

However, the comparison with PPO-Lag indeed raises some concerns in terms of the performance of FPO. First, it is clear that the initial performance of PPO-Lag reported in the paper was not well tuned. And as shown in the additional results in the rebuttal, by changing the value of the initial Lagrange multiplier and learning rate, the performance of PPO-Lag is almost comparable in many environments (11 out of 14 environments, and the average result is a little bit misleading because of the large performance gaps in a few velocity tasks). This raises a valid question that whether PPO-Lag can even outperform FPO through a careful selection of hyperparameters.

**Reviewer Scores:**

All reviewers may maintain their scores: reviewer 92qC's concern regarding the empirical performance was not fully addressed; reviewers KU9E and 6eAt were positive initially, and their concerns, which were mainly about clarifications and new baselines, have been well addressed in the rebuttal.

---

### Decision · Program_Chairs · 2026-01-26

Reject